# Scaling Up Bayesian Neural Networks with Neural Networks

**Zahra Moslemi**                                                                zmoslemi@uci.edu
*Department of Statistics*
*University of California*
*Irvine, CA, USA*

**Yang Meng**                                                                    mengy13@uci.edu
*Department of Statistics*
*University of California*
*Irvine, CA, USA*

**Shiwei Lan**                                                                   slan7@asu.edu
*School of Mathematical and Statistical Sciences*
*Arizona State University*
*Tempe, AZ, USA*

**Babak Shahbaba**                                                               babaks@uci.edu
*Department of Statistics*
*University of California*
*Irvine, CA, USA*

**Reviewed on OpenReview:** *https://openreview.net/forum?id=cD209UgOX7*

## Abstract

Bayesian Neural Networks (BNNs) offer a principled and natural framework for proper uncertainty quantification in the context of deep learning. They address the typical challenges associated with conventional deep learning methods, such as data insatiability, ad-hoc nature, and susceptibility to overfitting. However, their implementation typically either relies on Markov chain Monte Carlo (MCMC) methods, which are characterized by their computational intensity and inefficiency in a high-dimensional space, or variational inference methods, which tend to underestimate uncertainty. To address this issue, we propose a novel Calibration-Emulation-Sampling (CES) strategy to significantly enhance the computational efficiency of BNN. In this framework, during the initial calibration stage, we collect a small set of samples from the parameter space. These samples serve as training data for the emulator, which approximates the map between parameters and posterior probability. The trained emulator is then used for sampling from the posterior distribution at substantially higher speed compared to the standard BNN. Using simulated and real data, we demonstrate that our proposed method improves computational efficiency of BNN, while maintaining similar performance in terms of prediction accuracy and uncertainty quantification.

## 1 Introduction

In recent years, Deep Neural Networks (DNN) have emerged as the predominant driving force in the field of machine learning and are regarded as fundamental tools for many intelligent systems (Cheng et al., 2018; LeCun et al., 1998; Sze et al., 2017). While DNN have demonstrated significant success in prediction tasks, they often struggle with accurately quantifying uncertainty. Additionally, due to their vulnerability to overfitting, they can generate highly confident yet erroneous predictions (Su et al., 2019; Kwon et al., 2022). In recent years, there have been some attempts to address this issue. For example, the Ensemble

Deep Learning method (Lakshminarayanan et al., 2017) aggregates predictions from multiple models to improve reliability and uncertainty estimates. While these methods represent important progress in the right direction, developing a principled and computationally efficient framework for Uncertainty Quantification (UQ) within the context of deep learning remains a significant challenge and active area of research. This is especially important in domains where critical decisions, such as medical diagnostics, are involved. To address these issues, Bayesian Neural Networks (BNNs) (MacKay, 1992; Neal, 2012; Jospin et al., 2022) have emerged as an alternative to standard DNN, providing a more reliable framework within the field of machine learning. Their intrinsic ability to capture and quantify uncertainties in predictions establishes a robust foundation for decision-making under uncertainty. However, Bayesian inference in high-dimensional BNN poses significant computational challenges due to the inefficiency of traditional Markov Chain Monte Carlo (MCMC) methods. In fact, not only BNN, but almost all traditional Bayesian inference methods relying on MCMC techniques are known for their computational intensity and inefficiency when dealing with high-dimensional problems. Variational inference (Jordan et al., 1999) methods have been proposed to speed up computation by approximating the posterior distribution, but they tend to underestimate uncertainty (Minka, 2005). Consequently, researchers have proposed various approaches to expedite the inference process (Welling & Teh, 2011; Shahbaba et al., 2014; Ahn et al., 2014; Hoffman & Gelman, 2014; Beskos et al., 2017; Cui et al., 2016; Zhang et al., 2017; 2018; Li et al., 2019). Here, we focus on a state-of-the-art approach, called Calibration-Emulation-Sampling (CES) (Cleary et al., 2021), which has shown promising results in large-dimensional UQ problems such as inverse problems (Lan et al., 2022). CES involves the following three steps:

**(i)** Calibrate models to collect sample parameters and their corresponding expensive evaluation of posterior probability for the emulation step;

**(ii)** Emulate the parameter-to-posterior map using the samples from Step (i); and

**(iii)** Generate posterior samples using MCMC based on the trained emulator at substantially lower cost.

This framework allows for reusing expensive forward evaluations from parameters to posterior probability and offers a computationally efficient alternative to existing MCMC procedures.

The standard CES method (Cleary et al., 2021) focuses on UQ in inverse problems and uses Gaussian Process (GP) models for the emulation component. GP models have a well-established history of application in emulating computer models (Currin et al., 1988), conducting uncertainty analyses (Oakley & O'Hagan, 2002), sensitivity assessments (Oakley & O'Hagan, 2004), and calibrating computer codes (Kennedy & O'Hagan, 2002; Higdon et al., 2004). Despite their versatility, GP-based emulators are computationally intensive, with a complexity of $O(N^3)$ using the squared-exponential kernel, where $N$ is the sample size. Lower computational complexity can be achieved using alternative kernels (Lan et al., 2015) or various computational techniques (Liu et al., 2020; Bonilla et al., 2007; Gardner et al., 2018; Seeger et al., 2003). Nevertheless, scaling up GP emulators to high-dimensional problems remains a limiting factor. Furthermore, the prediction accuracy of GP emulators highly depends on the quality of the training data, emphasizing the importance of rigorous experimental design. To address these issues, Lan et al. (2022) proposed an alternative CES scheme called Dimension-Reduced Emulative Autoencoder Monte Carlo (DREAMC) method, which uses Convolutional Neural Networks (CNN) as emulator. DREAMC improves and scales up the application of the CES framework for Bayesian UQ in inverse problems from hundreds of dimensions (with GP emulation) to thousands of dimensions (with CNN emulation). Here, we adopt a similar approach and propose a new method, called Fast BNN (FBNN), for Bayesian inference in neural networks. We use DNN for the emulation component of our CES scheme. DNN has proven to be a powerful tool in a variety of applications and offers several advantages over GP emulation (Lan et al., 2022; Dargan et al., 2020). It is computationally more efficient and suitable for high-dimensional problems. The choice of DNN as an emulator enhances computational efficiency and flexibility.

Besides the computational challenges of building emulators, efficiently sampling from posterior distributions using these emulators also presents a significant challenge due to the high dimensionality of the target distribution. Traditional Metropolis-Hastings algorithms, typically defined on finite-dimensional spaces, encounter diminishing mixing efficiency as the dimensions increase (Gelman et al., 1997; Roberts & Rosenthal,

1998; Beskos et al., 2009). To overcome this inherent drawback, a novel class of dimension-independent MCMC methods has emerged, operating within infinite-dimensional spaces (Beskos, 2014; Beskos et al., 2009; 2011; Cotter et al., 2013; Law, 2014; Beskos, 2014; Beskos et al., 2017). More specifically, we use the Preconditioned Crank-Nicolson (pCN) algorithm. The most significant feature of pCN is its dimension robustness, which makes it well-suited for high-dimensional sampling problems. The pCN algorithm is well-defined, with non-degenerate acceptance probability, even for target distributions on infinite-dimensional spaces. As a result, when pCN is implemented on a real-world computer in large but finite dimension $N$, the convergence properties of the algorithm are independent of $N$. This is in strong contrast to schemes such as Gaussian random-walk Metropolis-Hastings and the Metropolis-adjusted Langevin algorithm, whose acceptance probability degenerates to zero as $N$ goes to infinity.

In summary, this paper addresses the critical challenges of UQ in high-dimensional BNN. By incorporating deep neural networks for emulation and leveraging the dimension-robust pCN algorithm for sampling, this research significantly enhances computational efficiency and scalability in Bayesian uncertainty quantification, offering a robust counterpart to DNN, and a scalable counterpart to BNN. Through extensive experiments, we demonstrate the feasibility and effectiveness of utilizing FBNN to accelerate Bayesian UQ in high-dimensional neural networks.

## 2 Related Methods

Various MCMC methods have been employed to explore complex probability distributions for Bayesian inference. In this section, we discuss some of the main MCMC algorithms related to our work. Additionally, we discuss a variety of state-of-the-art methods utilized in our numerical experiments, which extend beyond MCMC frameworks. These include Ensemble Deep Learning for Neural Networks (Perrone & Cooper, 1995), BNNs with Variational Inference (Jaakkola & Jordan, 2000), BNNs leveraging Lasso Approximation (MacKay, 1992), Monte Carlo Dropout (MC-Dropout) (Gal & Ghahramani, 2016), Stochastic Weight Averaging-Gaussian (SWAG) (Maddox et al., 2019), and Accelerated Hamiltonian Monte Carlo (HMC) (Zhang et al., 2017). These techniques offer a comprehensive spectrum for evaluating our FBNN model.

### 2.1 Hamiltonian Monte Carlo (HMC)

MCMC methods are designed for sampling from intractable probability distributions. The fundamental principle involves constructing a Markov chain whose equilibrium distribution coincides with the target distribution. Various algorithms exist for constructing such Markov chains, the simplest of which is the Metropolis-Hastings (MH) algorithm (Metropolis et al., 1953; Hastings, 1970), which relies on a random walk to explore the parameter space. Hamiltonian Monte Carlo (HMC) is a special case of the MH algorithm that incorporates Hamiltonian dynamics evolution and auxiliary momentum variables (Neal, 2011). Compared to using a Gaussian random walk proposal distribution in the MH algorithm, HMC reduces the correlation between successive sampled states by proposing moves to distant states that maintain a high probability of acceptance due to the approximate energy conserving properties of the simulated Hamiltonian dynamic. The reduced correlation means fewer Markov chain samples are needed to approximate integrals with respect to the target probability distribution for a given Monte Carlo error.

### 2.2 Stochastic Gradient Hamiltonian Monte Carlo (SGHMC)

As discussed earlier, HMC sampling methods provide a mechanism for defining distant proposals with high acceptance probabilities in a Metropolis-Hastings framework, enabling more efficient exploration of the state space than standard random-walk proposals. However, a limitation of HMC methods is the required gradient computation for simulation of the Hamiltonian dynamical system; such computation is infeasible in problems involving a large sample size. Stochastic Gradient Hamiltonian Monte Carlo (SGHMC) (Chen et al., 2014) addresses computational inefficiency by using a noisy but unbiased estimate of the gradient computed from a mini-batch of the data. SGHMC is an effective method for Bayesian inference, particularly when dealing with large datasets, as it leverages stochastic gradients and hyperparameter adaptation to efficiently explore high-dimensional target distributions.

---

**Algorithm 1** Preconditioned Crank-Nicolson (pCN) Algorithm

---

**Notation:**
  $k$: iteration index.
  $u^{(k)}$: state of the algorithm at iteration $k$.
  $\xi^{(k)} \sim \mathcal{N}(0, C)$: Gaussian noise with covariance matrix $C$ at iteration $k$.
  $\beta$: parameter controlling the proposal step size.
  $\Phi$: potential function related to the target probability distribution.
  $a(u^{(k)}, v^{(k)})$: acceptance probability for the proposed state $v^{(k)}$.

**Algorithm Steps:**
Initialize iteration counter $k \leftarrow 0$.
Choose an initial state $u^{(0)}$.
**repeat**
  Generate noise $\xi^{(k)}$ with $\xi^{(k)} \sim \mathcal{N}(0, C)$.
  Propose $v^{(k)} \leftarrow \sqrt{1 - \beta^2} u^{(k)} + \beta \xi^{(k)}$.
  Calculate acceptance probability $a(u^{(k)}, v^{(k)}) \leftarrow \min\{1, \exp(\Phi(u^{(k)}) - \Phi(v^{(k)}))\}$.
  **if** random number $\leq a(u^{(k)}, v^{(k)})$ **then**
    Set $u^{(k+1)} \leftarrow v^{(k)}$.
  **else**
    Set $u^{(k+1)} \leftarrow u^{(k)}$.
  **end if**
  Increment iteration counter $k \leftarrow k + 1$.
**until** a stopping criterion (fixed number of iterations or convergence) is met.

---

## 2.3 Random Network Surrogate–HMC (RNS–HMC)

Alternatively, we can reduce the computational cost of HMC by constructing surrogate Hamiltonians. As an example, the random network surrogate–HMC (RNS–HMC) method (Zhang et al., 2017) uses random non-linear bases to approximate posterior distributions. The goal is to explore and exploit the structure and regularity in parameter space for the underlying probabilistic model and construct an effective approximation of its geometric properties. To achieve this, RNS–HMC starts by identifying suitable bases that can capture the complex geometric properties of the parameter space. Through an optimization process, these bases are then used to form a surrogate function to approximate the expensive Hamiltonian dynamics. Unlike traditional HMC, which requires repeated evaluation of the model and its derivatives, RNS-HMC leverages the surrogate function to perform leapfrog integration steps, leading to a substantially lower computational cost. Later, Li et al. (2019) extended this idea by using a neural network to directly approximate the gradient of the Hamiltonian.

## 2.4 Preconditioned Crank-Nicolson (pCN)

Preconditioned Crank-Nicolson (Da Prato & Zabczyk, 2014) is a variant of MCMC that incorporates a preconditioning matrix for adaptive scaling (Cotter et al., 2013). It involves re-scaling and random perturbation of the current state, incorporating prior information. Despite the Gaussian prior assumption, the approach adapts to cases where the posterior distribution may not be Gaussian but is absolutely continuous with respect to an appropriate Gaussian density. This adaptation is achieved through the Radon-Nikodym derivative, connecting the posterior distribution with the dominating Gaussian measure, often chosen as the prior. The algorithmic foundation of pCN lies in using stochastic processes that preserve either the posterior or prior distribution. These processes serve as proposals for Metropolis-Hastings methods with specific discretizations, ensuring preservation of the Gaussian reference measure. The key steps of the pCN algorithm are outlined in Algorithm 1.

---

**Algorithm 2** Variational Inference in Bayesian Neural Networks (BNNs)

---

**Initialization:**

Choose an initial variational distribution $q_\theta(W)$ for the weights $W$ of BNN, parameterized by $\theta$.

Define the prior distribution $p(W)$ over the weights.

**while** not converged **do**

    **E-Step:** Estimate the Expectation of the log-likelihood over the variational distribution.

    Compute the gradient of the ELBO (Evidence Lower BOund) with respect to $\theta$, where

$$\text{ELBO}(\theta) = \mathbb{E}_{q_\theta(W)}[\log p(Y|X, W)] - \text{KL}[q_\theta(W)||p(W)]$$

    Here, $X$ and $Y$ are the inputs and outputs of the dataset, respectively, and KL denotes the Kullback-Leibler divergence between the variational distribution and the prior.

    **M-Step:** Maximize the ELBO with respect to $\theta$ using gradient ascent:

$$\theta \leftarrow \theta + \eta \nabla_\theta \text{ELBO}(\theta)$$

    where $\eta$ is the learning rate.

**end while**

**Output:** Variational distribution $q_\theta(W)$ approximating the posterior distribution $p(W|X, Y)$.

---

## 2.5 Variational Inference

The concept of variational inference has been applied in various forms to probabilistic models. The technique offers a way to approximate posterior distributions in Bayesian models (Jordan et al., 1999). The approximate distribution allows for a more feasible inference, especially for complex models like neural networks. In the context of BNNs, variational inference was brought into focus by Hinton & Van Camp (1993), which, while not explicitly termed as such in the modern sense, laid the groundwork for later developments. A more direct application of variational inference to BNNs was detailed in later works (e.g., Graves, 2011). More recently, Kingma & Welling (2014), Rezende et al. (2014) and Blundell et al. (2015) significantly contributed to popularizing and advancing the use of variational inference in deep learning and Bayesian neural networks through the introduction of efficient gradient-based optimization techniques.

Algorithm 2 succinctly shows the iterative process of optimizing the parameters of a variational distribution to approximate the posterior distribution of a BNN's weights. Through the alternation of expectation (E-Step) and maximization (M-Step) phases, it aims to minimize the difference between the variational distribution and the true posterior, leveraging the Evidence Lower Bound (ELBO) (Jordan et al., 1999) as a tractable surrogate objective function. This approach enables the practical application of Bayesian inference to neural networks, facilitating the quantification of uncertainty in predictions and model parameters.

## 2.6 Laplace Approximation

Previous studies have shown that in the context of BNNs, the Laplace approximation serves as an efficient method for approximating the posterior distribution over the network's weights (Arbel et al., 2023; Blundell et al., 2015). At the core of the Laplace approximation is the assumption that, around the loss function's minimum, the posterior distribution of the network's weights can be approximated by a Gaussian distribution. This is achieved by finding the mode of the posterior, called the Maximum A Posteriori (MAP; equivalent to the minimum of the loss function in the Bayesian framework), and then approximating the curvature of the loss surface at this point using the Hessian matrix (Liang et al., 2018). The inverse of this Hessian is used to define the covariance of the Gaussian posterior, thus simplifying the representation of uncertainty in the predictions. More specifically in BNNs, this approach can be used to approximate the posterior distribution of the weights given the observed data.

## 2.7  Monte Carlo Dropout

Monte Carlo (MC) Dropout (Gal & Ghahramani, 2016) was introduced as a Bayesian approximation method to quantify model uncertainty in deep learning. The core idea behind this method is to interpret dropout, a technique commonly used to prevent overfitting in neural networks, from a Bayesian perspective. Normally, dropout randomly disables a fraction of neurons during the training phase to improve generalization. However, when viewed through the Bayesian lens, dropout can be seen as a practical way to approximate Bayesian inference in deep networks. This approximation allows the network to estimate not just a single set of weights, but a distribution over them, enabling the model to express uncertainty in its predictions. The MC Dropout technique involves running multiple forward passes through the network with dropout enabled. Each forward pass generates a different set of predictions due to the random omission of neurons, leading to a distribution of outputs for a given input.

## 2.8  Stochastic Weight Averaging-Gaussian (SWAG)

Building on the idea of Stochastic Weight Averaging (SWA) (Izmailov et al., 2018; Maddox et al., 2019), SWAG approximates the distribution of model weights by a Gaussian distribution, leveraging the empirical weight samples collected from training. This approach allows for a more nuanced understanding of the model's uncertainty compared to SWA, which simply averages weights over the latter part of the training process. SWAG involves collecting a set of weights $\{W_i\}_{i=1}^{N}$ over the last $N$ epochs of training, where $W_i$ represents the weight vector at epoch $i$. The mean $\mu$ of the Gaussian distribution is then computed as the simple average of these weights: $\mu = \frac{1}{N}\sum_{i=1}^{N} W_i$. To capture the covariance of the weight distribution, SWAG calculates the empirical covariance matrix: $\Sigma = \frac{1}{N-1}\sum_{i=1}^{N}(W_i - \mu)(W_i - \mu)^T$. This formulation assumes a diagonal, or low-rank plus diagonal, approximation of the covariance matrix to maintain computational efficiency. The resulting Gaussian distribution, characterized by $\mu$ and $\Sigma$, can then be used for uncertainty estimation and prediction by sampling weights from this distribution and averaging the predictions of the resulting models.

## 3  Bayesian UQ for Neural Networks: Calibration-Emulation-Sampling

Standard neural networks (NN) typically consist of multiple layers, starting with an input layer, denoted as $l_0$, followed by a series of hidden layers $l_l$ for $l = 1, \ldots, m - 1$, and ending with an output layer $l_m$. In this architectural framework, comprising a total of $m + 1$ layers, each layer $l$ is characterized by a linear transformation, which is subsequently subjected to a nonlinear operation $g$, commonly referred to as an activation function (Jospin et al., 2022):

$$\begin{aligned} l_0 &= X, \\ l_l &= g_l\left(W_l l_{l-1} + b_l\right) \quad \text{for all } l \in \{1, \cdots, m-1\}, \\ l_m &= Y. \end{aligned} \tag{1}$$

Here, $\boldsymbol{\theta} = (\boldsymbol{W}, \boldsymbol{b})$ are the parameters of the network, where $\boldsymbol{W}$ are the weights of the network connections and $\boldsymbol{b}$ are the biases. A given NN architecture represents a set of functions isomorphic to the set of possible parameters $\boldsymbol{\theta}$. Deep learning is the process of estimating the parameters $\boldsymbol{\theta}$ from the training set $(\boldsymbol{X}, \boldsymbol{Y}) := \{(\boldsymbol{x_n}, \boldsymbol{y_n})\}_{n=1}^{N}$ composed of a series of input $\boldsymbol{X}$ and their corresponding labels $\boldsymbol{Y}$. Based on the training set, a neural network is trained to optimize network parameters $\boldsymbol{\theta}$ in order to map $\boldsymbol{X} \to \boldsymbol{Y}$ with the objective of obtaining the maximal accuracy (under certain loss function $L(\cdot)$). Considering the error, we can write NN as a forward mapping, denoted as $\mathcal{G}$, that maps each parameter vector $\boldsymbol{\theta}$ to a function that further connects $\boldsymbol{X}$ to $\boldsymbol{Y}$ with small errors $\varepsilon_n$:

$$\mathcal{G} : \Theta \to \boldsymbol{Y}^{\boldsymbol{X}}, \quad \boldsymbol{\theta} \mapsto \mathcal{G}(\boldsymbol{\theta}) \tag{2}$$

More specifically,

$$\mathcal{G}(\boldsymbol{\theta}) : \boldsymbol{X} \to \boldsymbol{Y}, \quad \boldsymbol{y_n} = \hat{\boldsymbol{y}}_n + \boldsymbol{\varepsilon_n}, \quad \hat{\boldsymbol{y}}_n = \mathcal{G}(\boldsymbol{x_n}; \theta), \quad \varepsilon \sim \boldsymbol{N(0, \Gamma)} \tag{3}$$

where $\varepsilon$ represents random noise capturing disparity between the predicted and actual observed values in the training data. Here, $\boldsymbol{Y}$ is a continuous random variable in regression problems, or a continuous *latent* variable in classification problems.

To train NN, stochastic gradient algorithms could be used to solve the following optimization problem:

$$\boldsymbol{\theta}^* \quad = \quad \arg\min_{\boldsymbol{\theta}\in\Theta} L(\boldsymbol{\theta}; \boldsymbol{X}, \boldsymbol{Y}) = \arg\min_{\boldsymbol{\theta}\in\Theta} L(\boldsymbol{Y} - \mathcal{G}(\boldsymbol{X}; \theta))$$

For example, the loss function $L(\boldsymbol{\theta}; \boldsymbol{X}, \boldsymbol{Y})$ can be defined in terms of the negative log-likelihood function $\Phi$ as follows:

$$\Phi(\boldsymbol{\theta}; \boldsymbol{X}, \boldsymbol{Y}) \quad = \quad \frac{1}{2}\|\boldsymbol{Y} - \mathcal{G}(\boldsymbol{X}; \boldsymbol{\theta})\|_{\Gamma}^2 \tag{4}$$

The point estimate approach, which is the traditional approach in deep learning, is relatively straightforward to implement with modern algorithms and software packages, but tends to lack proper uncertainty quantification (Guo et al., 2017; Nixon et al., 2019). To address this issue, stochastic neural networks, which incorporate stochastic components in the network, have emerged as a standard solution. This is performed by giving the network either stochastic activation functions or stochastic weights to simulate random samples for $\boldsymbol{\theta}$. The integration of stochastic components into neural networks allows for an extensive exploration of model uncertainty, which can be approached through Bayesian methods among others. It should be noted that not all neural networks that represent uncertainty are Bayesian or even stochastic; some employ deterministic methods to estimate uncertainty without relying on stochastic components or Bayesian inference (Lakshminarayanan et al., 2017; Sensoy et al., 2018). Bayesian neural networks (BNN) represent a subset of stochastic neural networks where Bayesian inference is specifically used for training, offering a rigorous probabilistic interpretation of model parameters (MacKay, 1992; Neal, 2012). The primary objective is to gain a deeper understanding of the uncertainty that underlies the specific process the network is modeling.

To design a BNN, we put a prior distribution over the model parameters, $p(\boldsymbol{\theta})$. By applying Bayes' theorem, the posterior probability can be written as:

$$p(\boldsymbol{\theta} \mid X, Y) \quad = \quad \frac{p(Y \mid X, \boldsymbol{\theta})\, p(\boldsymbol{\theta})}{\int_{\boldsymbol{\theta}} p(Y \mid X, \boldsymbol{\theta}')\, p(\boldsymbol{\theta}')\, d\boldsymbol{\theta}'} \tag{5}$$

$$\propto \quad p(Y \mid X, \boldsymbol{\theta})\, p(\boldsymbol{\theta}). \tag{6}$$

BNN is usually trained using MCMC algorithms. Because we typically have big amount of data, the likelihood evaluation tends to be expensive. One common approach to address this issue is subsampling, which restricts the computation to a subset of the data (see for example, Hoffman et al., 2010; Welling & Teh, 2011; Chen et al., 2014). The assumption is that there is redundancy in the data and an appropriate subset of the data can provide a good enough approximation of the information provided by the full data set. In practice, it is a challenge to find good criteria and strategies for an effective subsampling in many applications. Additionally, subsampling could lead to a significant loss of accuracy (Betancourt, 2015).

## 3.1 Fast Bayesian Neural Network (FBNN).

We propose an alternative approach that explores smoothness or regularity in parameter space, a characteristic common to most statistical models. Therefore, one would expect to find good and compact forms of approximation of functions (e.g., likelihood function) in parameter space. Sampling algorithms can use these approximate functions, also known as "surrogate" functions, to reduce their computational cost. More specifically, we propose using the CES scheme for high-dimensional BNN problems in order to bypass the expensive evaluation of original forward models and reduce the cost of sampling to a small computational overhead. Compared with MCMC methods, which require repeatedly evaluating the original (large) NN for the likelihood given the data, the proposed method builds a smaller NN emulator that bypasses the data (i.e., cuts out the middleman) by mapping the parameters directly to the likelihood function, thus avoiding costly evaluations. That is, the emulator is trained based on the parameter-likelihood pairs, which are collected through few iterations of the original BNN. In contrast to subsampling methods, this approach

---

**Algorithm 3** Fast Bayesian Neural Network (FBNN)

---

**Input:** Training set $\{(X_n, Y_n)\}_{n=1}^N$, Prior $p(\boldsymbol{\theta})$
**Output:** Posterior samples for model parameters
**procedure** FBNN($\{(X_n, Y_n)\}_{n=1}^N, p(\boldsymbol{\theta})$)
    **Calibration Step:**
    Initialize model parameters $\boldsymbol{\theta}$ using SGHMC
    Save posterior samples $\{\boldsymbol{\theta}_n^{(j)}\}_{j=1}^J$ and the corresponding $\{\mathcal{G}_{\boldsymbol{\theta}_n^{(j)}}(\boldsymbol{X_n})\}_{j=1}^J$ after a few iterations
    **Emulation Step:**
    Build an emulator of the forward mapping $\mathcal{G}^e$ based on $\{\boldsymbol{\theta}_n^{(j)}, \mathcal{G}_{\boldsymbol{\theta}_n^{(j)}}(\boldsymbol{X_n})\}_{j=1}^J$ using a DNN as the emulator
    **Sampling Step:** Run approximate MCMC, particularly $pCN$, based on the emulator to propose $\theta'$ from $\theta$.
**end procedure**

---

can handle computationally intensive likelihood functions, whether the computational cost is due to high-dimensional data or complex likelihood function (e.g., models based on differential equations). Additionally, the calibration process increases the efficiency of MCMC algorithms by providing a robust initial point in the high-density region. Algorithm 3 shows how our proposed method, called Fast Bayesian Neural Network (FBNN), combines the strengths of BNN in uncertainty quantification, SGHMC for efficient parameter calibration, and the pCN method for sampling. More details are provided in the following sections.

## 3.2 Calibration – Early stopping in Bayesian Neural Network

By "calibration" we mean collecting an optimal sample of parameters to build an emulator with a reasonable level of accuracy. This is aligned with traditional calibration goals of balancing accuracy and reliability, but within a new context. Here, the calibration step involves an early stopping strategy, aimed at collecting a targeted set of posterior samples without fully converging to the target distribution. More specifically, we use the Stochastic Gradient Hamilton Monte Carlo (SGHMC) algorithm for a limited number of iterations to collect a small set of samples. These samples include both the model parameters ($\boldsymbol{\theta}^{(j)}$) and the outputs predicted by the model ($\mathcal{G}(\boldsymbol{X}; \boldsymbol{\theta}^{(j)})$) for each sample $j$ out of a total of $J$ samples. The key focus of this training phase is not to obtain a precise approximation of the target posterior distribution, but rather collecting a small number of posterior samples as the "training data" for the subsequent emulation step. The SGHMC algorithm plays a crucial role in efficiently handling large datasets and collecting essential samples during the calibration step of the FBNN. Its ability to introduce controlled stochasticity in updates proves instrumental in preventing local minima entrapment, thereby providing a comprehensive set of posterior samples that capture the variability in the parameter space.

## 3.3 Emulation – Deep Neural Network (DNN)

The original forward mapping in BNN involves mapping input dataset $X$ to the response variable $Y$. For the likelihood evaluation using original forward mapping, it is necessary to calculate the likelihood $L(\boldsymbol{\theta}; X, Y)$ for each sample of model parameters. This means that with each iteration, when a new set of model parameters is introduced, the original forward mapping needs to be applied to generate output predictions, followed by the calculation of the likelihood. In general, this process can be very time-consuming. If, however, we have a small set of estimated model parameters along with their corresponding predicted outputs collected during the calibration step, an emulator can be trained to eliminate the intermediary step (passing through each data point), allowing us to map the parameters directly to the likelihood function. This leads to a computationally efficient likelihood evaluation. Therefore, to address the computational challenges of evaluating the likelihood with large datasets, we build an emulator $\mathcal{G}^e$ using the recorded pairs $\{\boldsymbol{\theta}^{(j)}, \hat{\boldsymbol{y}}^{(j)} = \mathcal{G}(\boldsymbol{X}; \boldsymbol{\theta}^{(j)})\}_{j=1}^J$ obtained during the calibration step. More specifically, these input-output pairs are used to train a DNN model as

an emulator $\mathcal{G}^e$ of the forward mapping $\mathcal{G}$:

$$\mathcal{G}^e(\boldsymbol{X};\boldsymbol{\theta}) = \text{DNN}(\boldsymbol{\theta},\mathcal{G}(\boldsymbol{X};\boldsymbol{\theta})) = F_{K-1} \circ \cdots \circ F_0(\boldsymbol{\theta}), \tag{7}$$

$$F_k(\cdot) = g_k(W_k \cdot +b_k) \in C\left(\mathbb{R}^{d_k},\mathbb{R}^{d_{k+1}}\right) \tag{8}$$

Given a DNN model where $\boldsymbol{\theta}$ represents the input and $\mathcal{G}(\boldsymbol{X};\boldsymbol{\theta})$ denotes the output, we set the dimensions as $d_0 = d$ and $d_K = D$, where $d$ represents the dimension of the input parameter vector $\boldsymbol{\theta}$, and $D$ represents the dimension of the output $\mathcal{G}(\boldsymbol{X};\boldsymbol{\theta})$. Here, the matrices $W_k$ are defined in the space $\mathbb{R}^{d_{k+1} \times d_k}$ and the vectors $b_k$ in $\mathbb{R}^{d_{k+1}}$. The functions $g_k$ act as (continuous) activation mechanisms. In the context of our numerical examples, the activation functions for the DNN emulator are selected to ensure that both the function approximations and their derived gradients have minimized errors. This involves a grid search over a predefined set of activation functions to ensure that the network efficiently approximates the target functions and their gradients.

After the emulator is trained, the log-likelihood can be approximated as follows:

$$L(\boldsymbol{\theta};X,Y) \approx L^e(\boldsymbol{\theta};X,Y) = L(Y - \mathcal{G}^e(X;\boldsymbol{\theta})) \tag{9}$$

By combining the approximate likelihood $L^e(\boldsymbol{\theta};X,Y)$ with the prior probability $p(\boldsymbol{\theta})$, an approximate posterior distribution can be obtained. Similarly, we could approximate the potential function using the predictions from DNN:

$$\Phi(\boldsymbol{\theta};\boldsymbol{X},\boldsymbol{Y}) \approx \Phi^e(\boldsymbol{\theta};\boldsymbol{X},\boldsymbol{Y}) = \frac{1}{2}\|\boldsymbol{Y} - \mathcal{G}^e(X;\boldsymbol{\theta})\|_\Gamma^2 \tag{10}$$

Building upon the foundational concepts of using a DNN emulator $\mathcal{G}^e$ for approximating the forward mapping function $\mathcal{G}$, we further elaborate on the implications and advantages of this approach for Bayesian inference, particularly in the context of handling large datasets and/or complex likelihood functions. The emulation step, which involves training the DNN emulator with input-output pairs $\{\boldsymbol{\theta}^{(j)},\mathcal{G}(\boldsymbol{X};\boldsymbol{\theta}^{(j)})\}$, serves as a critical phase where the emulator learns to mimic the behavior of the original model with high accuracy. The utilization of DNN emulator to approximate the likelihood function in Bayesian inference presents a significant computational advantage over the direct use of the original BNN likelihood. This advantage stems primarily from the inherent differences in computational complexity between evaluating the the likelihood with a DNN emulator – which takes a set of model parameters as input and yields predicted responses—and the original BNN model – which processes $\boldsymbol{X}$ as input to produce the response variable.

In the sampling stage, the computational complexity could be significantly reduced if we use $\Phi^e$ instead of $\Phi$ in the accept/reject step of MCMC. If the emulator is a good representation of the forward mapping, the difference between $\Phi^e$ and $\Phi$ would be small and negligible. Then, the samples by such emulative MCMC have the stationary distribution that closely follows the true posterior distribution. This approach not only ensures that the sampling process is computationally feasible, but also maintains the integrity of the stationary distribution, closely approximating the true posterior distribution with minimal discrepancy. The integration of DNN emulators into the Bayesian inference workflow thus presents a compelling solution to the computational challenges associated with evaluating likelihood functions in complex models.

### 3.4 Sampling – Preconditioned Crank-Nicolson (pCN)

In the context of the FBNN method, the sampling step is crucial for approximating the posterior distribution efficiently. The method employs MCMC algorithms based on a trained emulator to achieve full exploration and exploitation. However, challenges arise, especially in high-dimensional parameter spaces, where classical MCMC algorithms often exhibit increasing correlations among samples. To address this issue, the pCN method presented in Algorithm 1 has been used in our proposed framework as a potential solution. Unlike classical methods, pCN avoids dimensional dependence challenges, making it particularly suitable for scenarios like BNN models with a high number of weights to be inferred (Hairer et al., 2009).

As explained in section 2.4, the pCN approach minimizes correlations between successive samples, a critical feature for ensuring the representativeness of the samples collected. This characteristic is vital for FBNNs, as

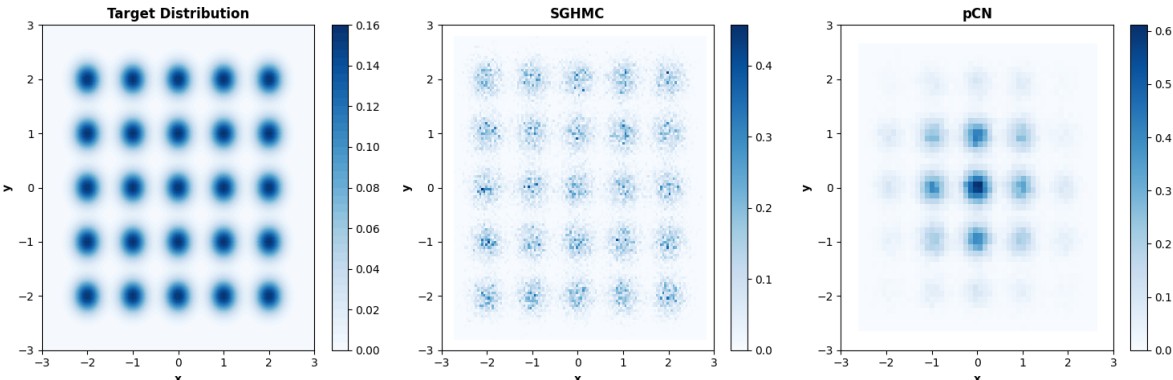

Figure 1: Sampling from a mixture of 25 Gaussians shown in (a) with 200k samples. SGHMC in (b) broadly explores the space, while pCN in (c) hones in on the high-density regions for precise mode capture.

it directly impacts the network's ability to learn from data and make robust predictions. The pCN method excels in traversing the parameter space with controlled perturbations, enhancing the algorithm's ability to capture the most probable configurations of model parameters. This focus on effective exploration around the mode contributes to a more accurate representation of the underlying neural network, ultimately improving model performance. In other words, the choice of pCN as the sampling method in FBNN is motivated by its tailored capacity to navigate and characterize the most probable regions of the parameter space. This choice reinforces the methodology's robustness and reliability, as pCN facilitates efficient sampling, leading to a more accurate and representative approximation of the posterior distribution.

To illustrate this, Figure 1 displays a simulation that contrasts the sampling mechanisms of SGHMC and pCN within a multimodal probability distribution. The task is to sample from a mixture of 25 Gaussian distributions, represented in panel (a), using a total of 200,000 samples. Here, the target distribution is multimodal with several distinct peaks (modes). Middle figure shows that SGHMC has explored the parameter space, although with a less concentrated sampling around the modes compared to the target distribution. This indicates that while SGHMC is effective at exploring the space, it may not capture the modes as tightly as the target distribution. In the right panel related to pCN sampler, the concentration of samples around the modes is much higher compared to SGHMC, which indicates that pCN is more effective at exploring around the modes of the distribution. Thus, we believe the combination of SGHMC and pCN in our proposed framework can complement each other for a more effective exploration of the parameter space.

### 3.5 Theoretical Foundations

In this section, we aim to quantify the error between the true potential function and its emulation in the context of the FBNN method. Let $\Omega = (0,1)^d$ and consider forward mappings in the Sobolev space $W^{n,p}(\Omega) := \{f \in L^p(\Omega) : D^\alpha f \in L^p(\Omega) \text{ for all } \alpha \in (\mathbb{N} \cup \{0\})^d \text{ with } |\alpha| \le n\}$ with $\|f\|_{n,p} := \left(\sum_{0 \le |\alpha| \le n} \|D^\alpha f\|_p^p\right)^{\frac{1}{p}}$, and $\|f\|_{n,\infty} := \max_{0 \le |\alpha| \le n} \|D^\alpha f\|_\infty$. Define the Sobolev-Slobodeckij norm for $0 < s < 1$: $\|f\|_{s,p} := \left(\|f\|_p^p + \int_\Omega \int_\Omega \frac{|f(x)-f(y)|^p}{|x-y|^{sp+d}} dx dy\right)^{\frac{1}{p}}$ and $\|f\|_{s,\infty} := \max\left\{\|f\|_\infty, \operatorname{ess\,sup}_{x,y \in \Omega} \frac{f(x)-f(y)}{|x-y|^s}\right\}$.

**Theorem 3.1.** *Let $1 \le p \le \infty$ and $0 \le s < 1$. Assume $\mathcal{G}_j(X; \cdot) \in W^{n,p}(\Omega) \cap L^\infty(\Omega)$ for $j = 1, \cdots, D$. For any $\epsilon \in (0, 1/2)$, there is a standard NN, $\mathcal{G}^e$, with ReLU activation functions such that*

$$\|\Phi - \Phi^e\|_{s,p} \le \epsilon. \tag{11}$$

*and the depth $K \le c \log(\epsilon^{-n/(n-s)})$, the number of weights and units $N \le c\epsilon^{-d/(n-s)} \log^2(\epsilon^{-n/(n-s)})$ with constant $c = c(d, n, p, s) > 0$.*

*Proof.* Note that we have

$$\Phi(\boldsymbol{\theta}) - \Phi^e(\boldsymbol{\theta}) = \frac{1}{2}[\langle \mathcal{G}(X;\boldsymbol{\theta}) - \mathcal{G}^e(X;\boldsymbol{\theta}), y - \mathcal{G}(X;\boldsymbol{\theta})\rangle_\Gamma + \langle y - \mathcal{G}^e(X;\boldsymbol{\theta}), \mathcal{G}(X;\boldsymbol{\theta}) - \mathcal{G}^e(X;\boldsymbol{\theta})\rangle_\Gamma]$$

Because $\mathcal{G}_j(X;\cdot) \in L^\infty((0,1)^d)$, there exists a constant $M > 0$ such that $\max_{1 \le j \le D} \|\mathcal{G}_j(X;\cdot)\|_\infty, \|y\| \le M$. For $\epsilon/(MD) > 0$, by Theorem 4.1 of Gühring et al. (2020), there exists a standard NN with ReLU activation functions and the depth $K$ and the number of weights and units as in the condition such that

$$\|\mathcal{G}_j(X;\cdot) - \mathcal{G}_j^e(X;\cdot)\|_{s,p} \le \epsilon/(MD), \quad j = 1, \cdots, D.$$

Therefore we have

$$\|\Phi - \Phi^e\|_{s,p} \le M \sum_{j=1}^{D} \|\mathcal{G}_j(X;\cdot) - \mathcal{G}_j^e(X;\cdot)\|_{s,p} \le \epsilon.$$

$\square$

Denote the Hellinger distance between densities as $d_H(\pi, \pi^e) = \int(\sqrt{\pi} - \sqrt{\pi^e})^e d\mu$. Then we describe how the emulation error propogates into the Hellinger error in the likelihood.

**Theorem 3.2.** *Let $\pi(\cdot;\boldsymbol{\theta}) \propto \exp(-\Phi(\cdot;\boldsymbol{\theta}))$ and $\pi^e(\cdot;\boldsymbol{\theta}) \propto \exp(-\Phi^e(\cdot;\boldsymbol{\theta}))$ denote the likelihood and its emulation, respectively. Suppose the conditions of Theorem 3.1 holds for $p = \infty$. Then we have*

$$d_H(\pi, \pi^e) \lesssim \epsilon. \tag{12}$$

*where $\epsilon$ satisfies the constraints in Theorem 3.1.*

*Proof.* Compute the Hellinger distance

$$2d_H^2(\pi, \pi^e) = \int(\sqrt{\pi} - \sqrt{\pi^e})^2 d\mu = \int \left[1 - \exp\left(\frac{1}{2}\Phi(y;\boldsymbol{\theta}) - \frac{1}{2}\Phi^e(y;\boldsymbol{\theta})\right)\right]^2 \pi(y;\boldsymbol{\theta})dy$$

$$\le \int \frac{C}{4}|\Phi(y;\boldsymbol{\theta}) - \Phi^e(y;\boldsymbol{\theta})|^2 \pi(y;\boldsymbol{\theta})dy \le C' \int \|\Phi - \Phi^e\|_\infty^2 \pi(y;\boldsymbol{\theta})dy \lesssim \epsilon^2.$$

where the last inequality is by Theorem 3.1. Taking square-root on both sides yields the conclusion. $\square$

Furthermore, given Gaussian prior for $\boldsymbol{\theta}$, $\pi_0$, we can characterize the discrepancy of posteriors with the original and emulated likelihoods in the following theorem.

**Theorem 3.3.** *Let $\hat{\pi}(\boldsymbol{\theta}) \propto \exp(-\Phi(y;\boldsymbol{\theta}))\pi_0(\boldsymbol{\theta})$ and $\hat{\pi}^e(\boldsymbol{\theta}) \propto \exp(-\Phi^e(y;\boldsymbol{\theta}))\pi_0(\boldsymbol{\theta})$ denote the posterior and the one with emulated likelihood, respectively. Suppose the conditions of Theorem 3.1 holds for $p = \infty$. Then we have*

$$d_H(\hat{\pi}, \hat{\pi}^e) \lesssim \epsilon. \tag{13}$$

*where $\epsilon$ satisfies the constraints in Theorem 3.1.*

*Proof.* By Bayes' theorem, we have

$$\hat{\pi}(\boldsymbol{\theta}) = \frac{1}{Z(y)}\exp(-\Phi(y;\boldsymbol{\theta}))\pi_0(\boldsymbol{\theta}), \quad 0 < Z(y) = \int_\Omega \exp(-\Phi(y;\boldsymbol{\theta}))\pi_0(\boldsymbol{\theta})d\boldsymbol{\theta} < +\infty$$

$$\hat{\pi}^e(\boldsymbol{\theta}) = \frac{1}{Z(y)}\exp(-\Phi^e(y;\boldsymbol{\theta}))\pi_0(\boldsymbol{\theta}), \quad 0 < Z^e(y) = \int_\Omega \exp(-\Phi^e(y;\boldsymbol{\theta}))\pi_0(\boldsymbol{\theta})d\boldsymbol{\theta} < +\infty.$$

Therefore we compute the Hellinger distance

$$
\begin{aligned}
d_H(\hat{\pi}, \hat{\pi}^e) &= \int_\Omega \left[ Z(y)^{-\frac{1}{2}} \exp\left(-\frac{1}{2}\Phi(y;\boldsymbol{\theta})\right) - Z^e(y)^{-\frac{1}{2}} \exp\left(-\frac{1}{2}\Phi^e(y;\boldsymbol{\theta})\right) \right]^2 \pi_0(d\boldsymbol{\theta}) \\
&\leq \frac{2}{Z(y)} \int_\Omega \left[ \exp\left(-\frac{1}{2}\Phi(y;\boldsymbol{\theta})\right) - \exp\left(-\frac{1}{2}\Phi^e(y;\boldsymbol{\theta})\right) \right]^2 \pi_0(d\boldsymbol{\theta}) + 2|Z(y)^{-\frac{1}{2}} - Z^e(y)^{-\frac{1}{2}}|^2 Z^e(y) \\
&\leq 2 \int_\Omega \left[ 1 - \exp\left(\frac{1}{2}\Phi(y;\boldsymbol{\theta}) - \frac{1}{2}\Phi^e(y;\boldsymbol{\theta})\right) \right]^2 \hat{\pi}(d\boldsymbol{\theta}) + C|Z(y) - Z^e(y)|^2 \\
&\leq \int \frac{C'}{2}|\Phi(y;\boldsymbol{\theta}) - \Phi^e(y;\boldsymbol{\theta})|^2 \hat{\pi}(d\boldsymbol{\theta}) + C \int_\Omega |\exp(-\Phi(y;\boldsymbol{\theta})) - \exp(-\Phi^e(y;\boldsymbol{\theta}))|^2 \pi_0(d\boldsymbol{\theta}) \\
&\leq \frac{C'}{2}\|\Phi - \Phi^e\|_\infty^2 + C''\|\Phi - \Phi^e\|_\infty^2 \lesssim \epsilon^2
\end{aligned}
$$

where the last inequality is by Theorem 3.1. Taking square-root on both sides yields the conclusion. $\qquad\square$

## 4 Numerical Experiments

### 4.1 Setup

We demonstrate the effectiveness of our method on eleven synthetic and real-world datasets, comparing it against a comprehensive selection of baseline approaches.

**Datasets.** We experiment on a series of regression and classification problems. Detailed information regarding these datasets, including the number of features and datapoints in each, and the number of parameters used in the main FBNN model for each dataset, is outlined in Table 1. We have also included the details of the DNN Emulator architecture for each dataset in Table 2.

**Baseline Methods.** We present empirical evidence comparing our CES method against a broad array of baseline approaches including two baseline BNN methods equipped with the SGHMC and pCN samplers (shown as BNN-SGHMC and BNN-pCN), and BNN architectures incorporating Variational Inference, Lasso Approximation, MC-Dropout, SWAG, and RNS-HMC. Detailed information about these methods was provided in Section 2. We also include the results from DNN, which does not provide uncertainty quantification, but serves as a reference point. Moreover, we provide the results of Deep Ensembles, which consist of multiple DNNs, each initialized with different random seeds. We refer to this method as Ensemble-DNN. Although the Ensemble-DNN approach allows for parallelization, it falls short in providing a probabilistic framework for analysis, a significant advantage offered by our CES method.

As discussed earlier, one of the distinctive features of our main FBNN model, more specifically shown as FBNN (SGHMC-pCN), lies in its strategic integration of the SGHMC sampler during the calibration step and the pCN algorithm during the sampling step. This combination is carefully chosen to harness the complementary strengths of these two sampling methods. Further expanding our exploration, we introduce three additional FBNN models: FBNN (pCN-SGHMC), where pCN is employed in the calibration step and SGHMC in the sampling step; FBNN (pCN-pCN), where pCN is used in both steps; and FBNN (SGHMC-SGHMC), where SGHMC is used in both calibration and sampling steps.

Throughout these experiments, we collect 2000 posterior samples for the BNN-SGHMC and BNN-pCN, with samples being collected at each iteration. In contrast, for the FBNN methods, we use a small number (200) of samples from either BNN-SGHMC or BNN-pCN (depending on the specific FBNN model) along with the corresponding predicted outputs during the calibration step. These 200 samples serve as the "training data" for the emulator. Moreover, we evaluate the efficacy of utilizing only the initial 200 samples from the BNN-SGHMC model across all the datasets. This was done to demonstrate the necessity of collecting more samples, either using the original BNN or employing the FBNN method, rather than relying our inference on a limited number of initial samples.

Table 1: Description of various datasets used to evaluate the overall performance of our proposed approach against state-of-the-art baseline methods.

| Task | Dataset | # Datapoints | # Features | # FBNN parameters |
|---|---|---|---|---|
| Regression | Boston Housing | 506 | 13 | 3,009 |
| | Wine Quality | 1,599 | 11 | 241 |
| | Alzheimer | 185,831 | 56 | 33,345 |
| | Year Prediction MDS | 515,345 | 90 | 81,901 |
| | Simulation | 5,000,000 | 1,000 | 52,832 |
| Classification | Adult | 40,434 | 14 | 2,761 |
| | Mnist | 70,000 | 784 | 3,961 |
| | Alzheimer | 185,831 | 56 | 33,345 |
| | celebA | 202,599 | 39 | 1,521 |
| | SVHN | 630,420 | 3072 | 171,313 |
| | Simulation | 5,000,000 | 1,000 | 52,832 |

Table 2: Description of various datasets and their corresponding DNN Emulator architectures (Droupout layers have been used on input layer and first hidden layer)

| Task | Dataset | #Hidden Layers | # Neurons per Layer | Activation Functions | #Epochs | Dropout Rate |
|---|---|---|---|---|---|---|
| Regression | Boston Housing | 2 | 3,32 | ReLU | 1000 | 0.7 |
| | Wine Quality | 2 | 3,32 | ReLU | 1000 | 0.5 |
| | Alzheimer | 2 | 4,64 | ReLU | 1000 | 0.5 |
| | Year Prediction | 2 | 4,64 | ReLU | 1000 | 0.5 |
| | Simulation | 3 | 8,64,32 | ReLU | 1000 | 0.5 |
| Classification | Adult | 2 | 4,32 | ReLU | 1000 | 0.5 |
| | Mnist | 2 | 3,64 | ReLU | 1000 | 0.5 |
| | Alzheimer | 2 | 4,64 | ReLU | 1000 | 0.5 |
| | celebA | 2 | 3,32 | ReLU | 1000 | 0.5 |
| | SVHN | 3 | 4,64,32 | ReLU | 1000 | 0.7 |
| | Simulation | 3 | 8,64,32 | ReLU | 1000 | 0.5 |

It is also crucial to highlight that the BNN-SGHMC and BNN-pCN models are trained from a randomly chosen initial point for the MCMC sampling process. On the other hand, in the FBNN methods, we employ the set of posterior samples collected during the last iteration of the calibration step as the starting point for the subsequent MCMC sampling.

**Metrics.** To thoroughly assess the performance and effectiveness of each method, we use a range of key metrics. These include Mean Squared Error (MSE) for regression tasks (Figure 2) and Accuracy for classification tasks (Figure 3). We also evaluate the models based on their computational cost, and various statistics related to the Effective Sample Size (ESS) of model parameters. These statistics include the minimum, maximum, and median ESS, as well as the minimum ESS per second. We also quantify the amount of speedup, denoted as "spdup", a metric that compares the minimum ESS per second of each model with that of BNN-SGHMC as the benchmark (Figure 4). Analysing spdup is crucial as it provides a comparative

measure of efficiency, highlighting the model's capability to achieve high-quality parameter sampling with lower computational resource utilization relative to the benchmark BNN-SGHMC.

The effective sample size takes the autocorrelation among the consecutive samples into account. While we can reduce autocorrelation using the thinning strategy, this leads to a higher computational time for the same number of samples. Our spdup metric allows for a fair comparison of sampling algorithms (regardless of what thinning strategy used) by taking both autocorrelation and computational cost into account.

For UQ in regression cases, we evaluate the Coverage Probability (CP) set at 95%. In addition, we construct 95% Credible Intervals (CI) by the prediction results of Bayesian models, along with the average true output, to illustrate UQ in regression problems. For classification problems, we use Expected Calibration Error (ECE) and Reliability Diagrams to evaluate UQ. ECE addresses model calibration, aiming for accurate uncertainty estimates, while reliability diagrams offer a visual summary of probabilistic forecasts.

Figures 2, 3, and 4 summarize our results. More detailed results are provided in Tables A1 and A2.

### 4.2   Regression Tasks

We first evaluate our proposed method using a set of simulated and real regression problems.

**Simulated Data.**   We begin our empirical evaluation by using simulated data. To this end, we utilize the `make_regression` function from the `sklearn.datasets` package to generate a dataset consisting of 5,000,000 observations and 1,000 predictors.

Figure 5 compares the true and emulated log likelihood functions associated with posterior samples collected using BNN-SGHMC. The emulated values are based on the FBNN (SGHMC-pCN) model. As we can see, the two functions are similar, indicating that the emulator provides a reasonable approximation of the true target distribution.

Figure 2 compares the MSE among all models, showing that while the DNN method achieves the lowest MSE at 0.71, the FBNN (SGHMC-pCN) model provides a similar performance. Notably, among all the FBNN variants, FBNN (SGHMC-pCN) provides the highest CP at 92.2%, demonstrating a level of calibration comparable to that of the BNN model. The Ensemble-DNN demonstrates comparable performance to FBNN (SGHMC-pCN) in terms of CP, yet it operates at a pace three times slower.

Examining the efficiency of sample generation, all FBNN variants have relatively higher ESS per second compared to all the other BNN models, except for BNN-RNS-HMC. Among all the models, FBNN (SGHMC-pCN) has the highest min ESS per second at 0.043. Figure 4 indicates our model provides the highest speedup (16.33) compared to BNN-SGHMC as the baseline model, highlighting our method's computational efficiency. Considering these results, FBNN (SGHMC-pCN) emerges as a strong approach with a good balance between predictive accuracy, computational efficiency, and uncertainty quantification, making it the overall best option for Bayesian deep learning.

Figure 6a shows the estimated mean and prediction uncertainty for both BNN-SGHMC and FBNN (SGHMC-pCN) models, alongside the smoothed average and 95% interval for the true output. For clarity and conciseness within our figures, we have employed Principal Component Analysis (PCA) and used the first principal component to transform the original data into a one-dimensional representative feature (x-axis in Figure 6). As we can see, BNN and FBNN have very similar credible intervals. This consistency in credible interval bounds is significant for UQ, indicating that both models effectively and almost equally quantify uncertainty in their predictions.

**Wine Quality Data.**   As the first real dataset for the regression task, we use the Wine Quality data (Cortez et al., 2009). This dataset contains various physicochemical properties of different wines, while the target variable is the quality rating. The performance of FBNN (SGHMC-pCN) indicates a well-balanced approach, making it superior to the other models for several reasons. Firstly, it achieves a competitively low MSE of 0.52, comparable to other high-performing models like BNN-SGHMC and SWAG, but it surpasses them in terms of speedup. Moreover, FBNN (SGHMC-pCN) exhibits a robust predictive performance,

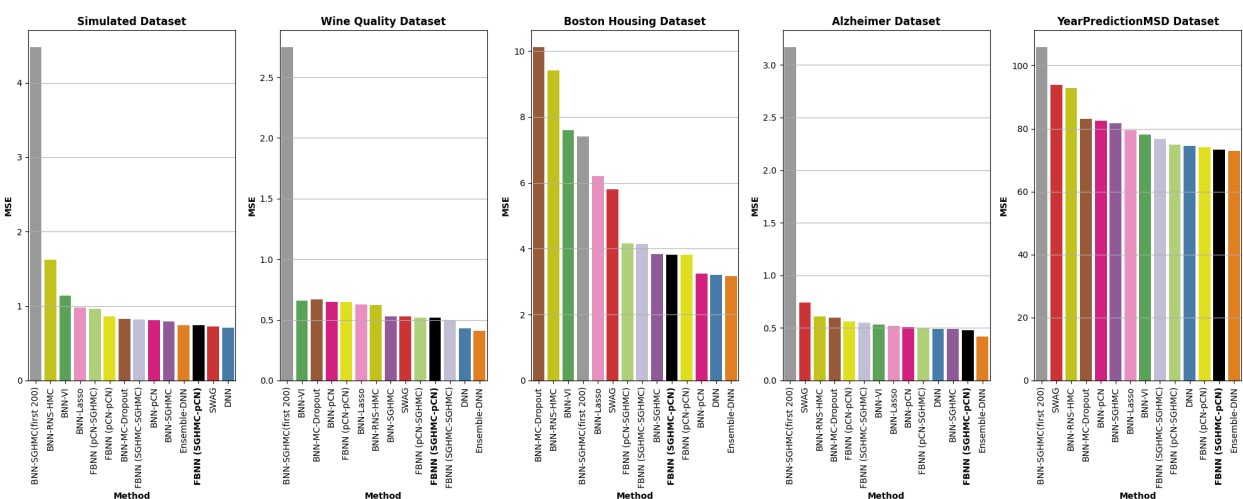

Figure 2: A comprehensive comparison of the MSE for various BNN methods across five regression datasets. Each subplot corresponds to a dataset and the color-coded bars represent the distinct methodologies evaluated. The main FBNN method, called FBNN (SGHMC-pCN), highlighted in bold, shows among the lowest values for MSE among all datasets.

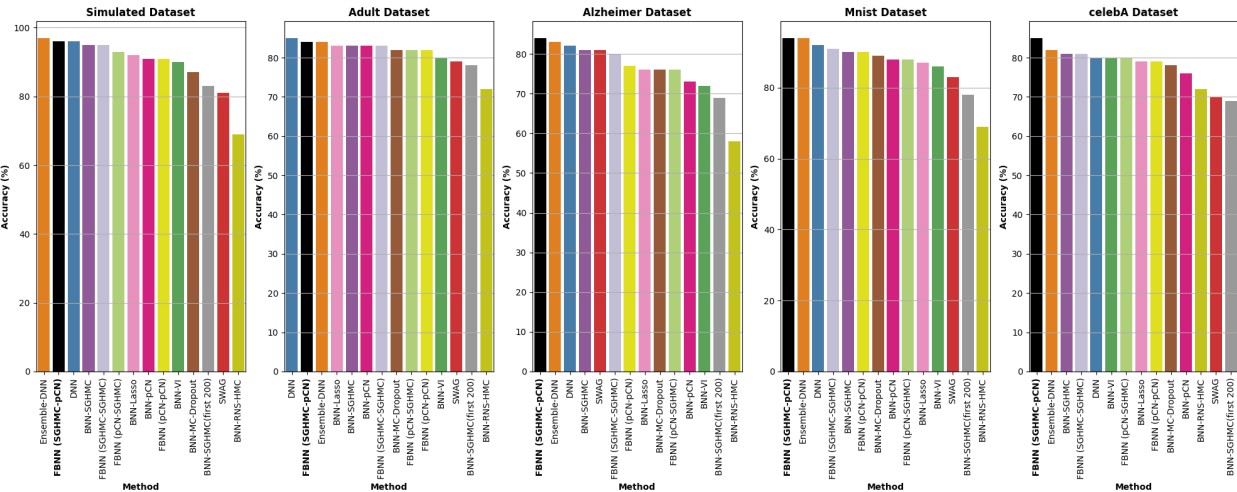

Figure 3: A comprehensive comparison of the Accuracy for various BNN methods across five classification datasets. Each subplot corresponds to a dataset and the color-coded bars represent the distinct methodologies evaluated. The main FBNN method, FBNN (SGHMC-pCN), highlighted in bold, demonstrates superior performance by achieving the highest accuracy in four out of the five datasets examined and the second highest in one dataset.

with a competitively high CP. Figure 6b shows the prediction mean and 95% CI BNN-SGHMC and FBNN (SGHMC-pCN), as well as the smoothed average and 95% interval for the true output.

**Boston Housing Data.** The Boston housing dataset was collected in 1978 (Harrison Jr & Rubinfeld, 1978). Each of the entries present aggregated data for homes from various suburbs in Boston. For this dataset, FBNN (SGHMC-pCN) stands out with a notable balance between MSE (3.82), CP (81.1%), and computational efficiency, completing the task in just 91 seconds. This model significantly outperforms all the other models in terms of speedup (11.94), showcasing its effectiveness in sampling. Figure 6c shows the 95% CIs and mean predictions of both BNN-SGHMC and FBNN (SGHMC-pCN). The FBNN (SGHMC-pCN)

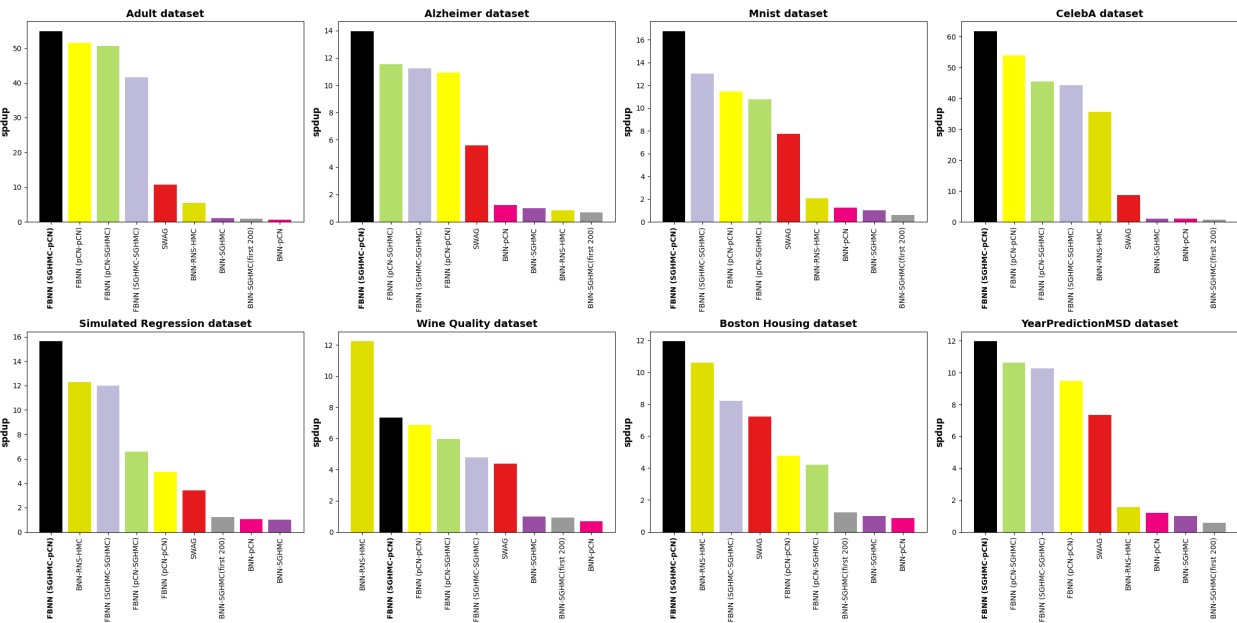

Figure 4: Comparative analysis of speedup (spdup) for various Bayesian Neural Network (BNN) methods across tested datasets. Methods are ordered by efficiency within each dataset, highlighting the impact of model characteristics on sampling performance. Notably, "FBNN (SGHMC-pCN)" achieves the highest speedup among all datasets except for the Wine Quality dataset, where it ranks as the second highest, underscoring its exceptional efficiency in diverse analytical contexts.

model, in particular, displays well-calibrated uncertainty quantification, mirroring the performance of the BNN models, implying that its probabilistic predictions capture the model uncertainty.

**Alzheimer Data.** Next, we analyze the data from the National Alzheimer's Coordinating Center (NACC), which is responsible for developing and maintaining a database of patient information collected from the Alzheimer disease centers (ADCs) funded by the National Institute on Aging (NIA) (Beekly et al., 2004). The NIA appointed the ADC Clinical Task Force to determine and define an expanded, standardized clinical data set, called the Uniform Data Set (UDS). The goal of the UDS is to provide ADC researchers a standard set of assessment procedures to identify Alzheimer's disease (Beekly et al., 2007). We have used 56 key features for our analysis. These features were carefully selected to represent a wide spectrum of variables relevant to Alzheimer's disease diagnosis, including functional abilities, brain morphometrics, living situations, and demographic information (Ren et al., 2022). For the regression case, the goal is to predict Left Hippocampus Volume, a critical marker in the progression of the disease (van der Flier & Scheltens, 2009), as a function of other variables. For this dataset, Figure 2 shows that the FBNN (SGHMC-pCN) model stands out for its balanced performance, recording the second lowest MSE at 0.48 and a relatively high CP at 91.6%. It shows a considerable improvement in computational efficiency, evidenced by a speedup factor of 22 times compared to BNN-SGHMC as the baseline BNN model.

**Year Prediction MSD Data.** For this data, the goal is to predict the release year of a song from audio features. Songs are mostly western, commercial tracks ranging from 1922 to 2011, with a peak in the year 2000s (Bertin-Mahieux, 2011). In the context of the YearPredictionMSD dataset, FBNN (SGHMC-pCN) showcases its superiority over other models by achieving a good balance between accuracy, computational efficiency, and effective uncertainty quantification. With an MSE of 73.41, close to that of Ensemble-DNN, and CP of 92.23% it outperforms most other models. Moreover, the computational efficiency of FBNN (SGHMC-pCN) is highlighted by its speedup factor of 11.98 (Figure 4) over the baseline model BNN-SGHMC.

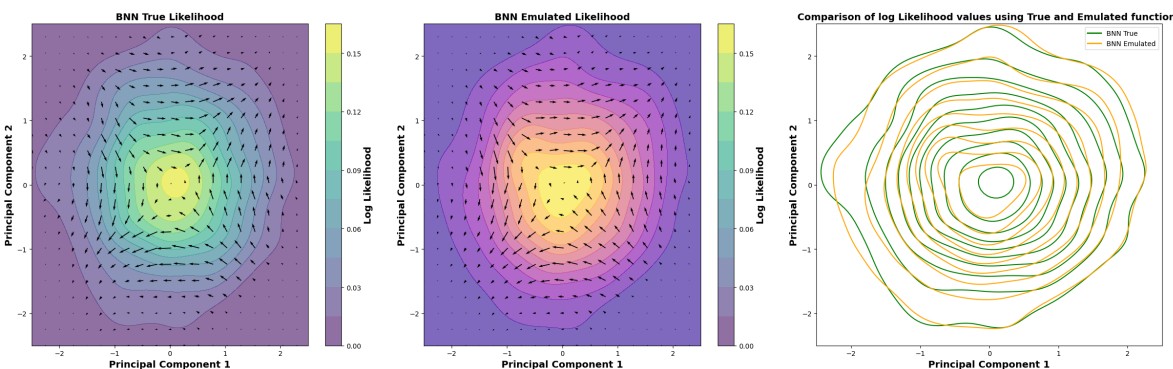

Figure 5: Evaluating the performance of the emulator. The plot contrasts log likelihood values obtained using the true likelihood function against those derived using the emulated likelihood function. The x-axis and y-axis represent the first and second principal components of the model parameters based on the MCMC samples obtained in our simulation study.

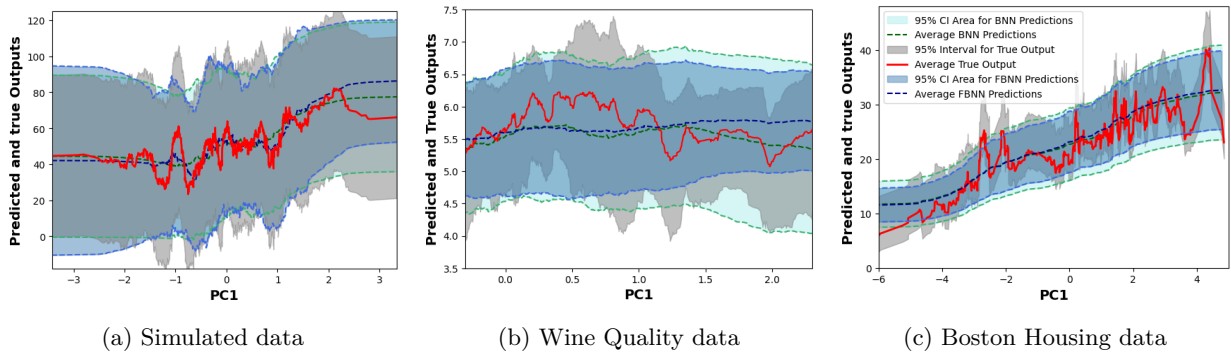

(a) Simulated data      (b) Wine Quality data      (c) Boston Housing data

Figure 6: Comparative analysis of predictive credible intervals and mean predictions for regression tasks. For each dataset, the 95% CI for BNN predictions and FBNN predictions are shown as shaded areas. The average predictions from BNN and FBNN are represented with dashed lines. Additionally, the 95% CI for the true output as ground truth and the smoothed average true output are plotted as solid lines. The x-axis shows the first principal component of the predictors.

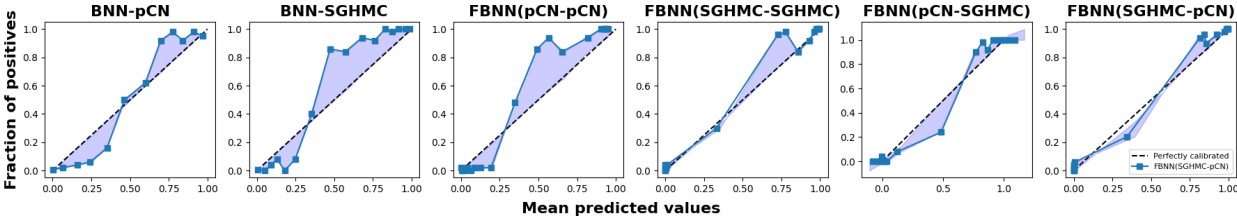

Figure 7: Reliability Diagrams for Simulated dataset in classification task. These diagrams incorporate equal frequency binning.

## 4.3 Classification Tasks

Next, we evaluate our method based on a set of simulated and real classification problems. The results are summarized in Figures 3 and 4. More details results are provided in Table A2.

**Simulated Data.** As before, we start with a simulated dataset with a binary outcome. For this, we use the `make_classification` function from the `sklearn.datasets` package to generate a dataset consisting of

5,000,000 observations and 1000 predictors. Accuracy comparison in Figure 3 shows FBNN (SGHMC-pCN), DNN, and Ensemble-DNN exhibit comparable performance and outperform other models. While BNN-RNS-HMC achieves the highest speedup, it significantly underperforms in terms of accuracy. In contrast, FBNN (SGHMC-pCN) provides the second-highest speedup at 32.00, showcasing its computational efficiency relative to BNN-SGHMC. Furthermore, it maintains the second highest accuracy rate of 96%, indicating an optimal balance between computational efficiency and accuracy.

For this example, BNN-MC-Dropout has the lowest ECE value compared to other methods (see Appendix), but it also have a lower accuracy rate. Among the FBNN variants, FBNN (SGHMC-pCN) presents a low ECE, closely aligning with the ECE value of BNN-MC-Dropout, while providing an accuracy rate similar to DNN. Moreover, as illustrated in Figure 7, the FBNN variants, particularly FBNN (SGHMC-pCN), appear to be better calibrated across most probability ranges except at the highest probabilities, suggesting a more reliable UQ. Note that in these figures, a model with a reliability curve that closely follows the diagonal line is considered better calibrated, meaning its predicted probabilities are more aligned with actual outcomes.

**Adult Data.** Next, we use the Adult dataset (Becker & Kohavi, 1996), where the classification task involves predicting whether an individual will earn more or less than $50,000 per year. Figure 3 demonstrates that all the methods achieve comparable accuracy rates, although DNN, Ensemble-DNN, and FBNN(SGHMC-pCN) have the best performance. FBNN (SGHMC-pCN) also stands out as the most computationally efficient method for uncertainty quantification, with a speedup value of 54.91 relative to the baseline BNN approach. A low ECE value for the FBNN (SGHMC-pCN) model signifies its superior performance in terms of uncertainty quantification.

**Alzheimer Data.** We have used the same NACC dataset we previously discussed, but this time as a classification problem. Here, our objective is to predict cognitive status, classifying individuals as either cognitively unimpaired (healthy controls, HC), labeled as class 0, or as having mild cognitive impairment (MCI) due to Alzheimer's disease (AD) or dementia due to AD, labeled as class 1.

Achieving the highest accuracy of 84% at a relatively low computational cost, FBNN (SGHMC-pCN) surpasses all other models in correctly identifying Alzheimer's disease, making it the most reliable model among those tested. This model not only surpasses the accuracy of the standard DNN and Ensemble-DNN models but also offers a balance between computational efficiency and high accuracy. FBNN (SGHMC-pCN) demonstrates the highest sampling efficiency (speedup of 11), indicating it can achieve high accuracy with a lower computational cost compared to other Bayesian models. Moreover, for the FBNN model implemented on the Alzheimer dataset, the ECE value is low, and the reliability curve closely tracks the diagonal line.

**MNIST Dataset.** The MNIST dataset is commonly used as a benchmark dataset for the hand-written digit classification task (Deng, 2012). Among the various models evaluated on the MNIST dataset, FBNN (SGHMC-pCN) stands out as the optimal choice, demonstrating exceptional performance across multiple metrics. It achieves the highest accuracy of 94%, matching the top performance of Ensemble-DNN but with significantly improved efficiency and effectiveness in uncertainty quantification. It exhibits a substantial speedup of 16.77, and the lowest ECE at 0.241, suggesting that it not only provides highly accurate predictions but also reliably estimates the uncertainty associated with these predictions.

**CelebA Dataset.** CelebA (Liu et al., 2015) is an image dataset of celebrity faces annotated with 40 attributes including gender, hair color, age, smiling, etc. The task is to predict hair color, which is either blond $Y = 1$ or non-blond $Y = 0$. The FBNN (SGHMC-pCN) model stands out among the alternative methods, showcasing its superiority through several key performance metrics. It achieves the highest accuracy of 85%, and the highest speedup factor of 61.84, indicating an exceptional balance between computational efficiency and performance. The model achieves the lowest ECE of 0.493, indicating reliability in its predictive uncertainty.

**SVHN dataset.** The Street View House Numbers (SVHN) dataset (Netzer et al., 2011) includes labelled real-world images of house numbers taken from Google Street View. The images are 32x32 pixels in size and have three color channels (RGB). The goal is to classify digit images into 10 classes. The results demonstrate

the superiority of FBNN (SGHMC-pCN) in terms of accuracy and computational efficiency. FBNN (SGHMC-pCN) achieved an accuracy of 96%, the second highest among all models. The speedup compared to the baseline BNN-SGHMC method was 14.99 times, the highest speedup value recorded. Additionally, the low ECE of 0.203 demonstrates better uncertainty quantification than most other methods, including BNN-VI, BNN-MC-Dropout, and SWAG.

## 5 Conclusion

In this paper, we have proposed an innovative CES framework called FBNN, specifically designed to enhance the computational efficiency and scalability of BNN for high-dimensional data. Our primary goal is to provide a robust solution for uncertainty quantification in high-dimensional spaces, leveraging the power of Bayesian inference while mitigating the computational bottlenecks traditionally associated with BNN.

In our numerical experiments, we have successfully applied several variants of FBNN, including different configurations with BNN, to regression and classification tasks on both synthetic and real datasets. Remarkably, the FBNN variant incorporating SGHMC for calibration and pCN for sampling, denoted as FBNN (SGHMC-pCN), not only matches the predictive accuracy of traditional BNN but also offers substantial computational advantages. More specifically, our numerical experiments across various regression and classification tasks consistently demonstrate the superiority of the FBNN (SGHMC-pCN) method over traditional BNNs and DNNs. In regression tasks, FBNN (SGHMC-pCN) demonstrates a competitive MSE while significantly enhancing computational efficiency, achieving notable speedups compared to baseline models. This efficiency does not come at the expense of accuracy, as evidenced by the competitive MSE values and robust uncertainty quantification metrics. In classification tasks, FBNN (SGHMC-pCN) stands out by achieving high accuracy rates and low ECE values, which indicate reliable uncertainty quantification.

The superior performance of FBNN (SGHMC-pCN) can be attributed to the complementary strengths of SGHMC and pCN. SGHMC excels at broad exploration of the parameter space, providing an effective means for understanding the global structure during the calibration step. On the other hand, pCN is adept at efficient sampling around modes, offering a valuable tool for capturing local intricacies in the distribution during the final sampling step. By combining these samplers within the FBNN framework, we achieve a balanced approach between exploration (calibration with SGHMC) and exploitation (final sampling with pCN).

Future work could involve extending our method to more complex problems (e.g., spatiotemporal data) and complex network structures (e.g., graph neural networks). Additionally, future research could focus on improving the emulation step by optimizing the DNN architecture. Finally, our method could be further improved by embedding the sampling algorithm in an adaptive framework similar to the method of Zhang et al. (2018).

## Acknowledgements

The authors thank the Editor, Action Editor, and anonymous reviewers for their insightful suggestions and constructive feedback, which significantly improved the article. This work was supported by NSF grants NCS-FR-2319618 and DMS-2134256.

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

## Appendix A: Comparing Various Deep Learning Techniques for Regression and Classification Problems

Table A1: Performance of various deep learning methods based on regression problems. For ESS, minimum, median, and maximum values are provided in parenthesis.

| Dataset | Method | MSE | CP | Time (s) | ESS | minESS/s | spdup |
|---|---|---|---|---|---|---|---|
| Simulated | DNN | 0.71 | - | 3531 | - | - | - |
| Dataset | Ensemble-DNN | 0.74 | 91.3% | 14633 | - | - | - |
| | BNN-VI | 1.14 | 88.8% | 8408 | - | - | - |
| | BNN-Lasso | 0.98 | 87.4% | 6941 | - | - | - |
| | BNN-MC-Dropout | 0.83 | 93.4% | 2861 | - | - | - |
| | BNN-SGHMC | 0.79 | 91.6% | 41281 | (109, 829, 1642) | 0.002 | 1.00 |
| | BNN-SGHMC(first 200) | 4.48 | 92.8% | 3970 | (21, 66, 162) | 0.005 | 2.01 |
| | SWAG | 0.73 | 90.0% | 13488 | (93, 1163, 1542) | 0.006 | 2.61 |
| | BNN-RNS-HMC | 1.62 | 87.5% | 6046 | (126, 948, 1510) | 0.021 | 7.89 |
| | BNN-pCN | 0.81 | 89.3% | 42523 | (107, 844, 1533) | 0.002 | 0.95 |
| | FBNN (pCN-SGHMC) | 0.96 | 79.6% | 4512 | (132, 1241, 1615) | 0.029 | 11.08 |
| | FBNN (pCN-pCN) | 0.86 | 85.2% | 4631 | (94, 975, 1621) | 0.020 | 7.69 |
| | FBNN (SGHMC-SGHMC) | 0.82 | 90.1% | 4497 | (137, 1236, 1638) | 0.030 | 11.53 |
| | FBNN (SGHMC-pCN) | 0.74 | 92.2% | 4312 | (186, 912, 1606) | 0.043 | 16.33 |
| Wine | DNN | 0.43 | - | 26 | - | - | - |
| Quality | Ensemble-DNN | 0.41 | 47.4% | 137 | - | - | - |
| | BNN-VI | 0.66 | 39.5% | 28 | - | - | - |
| | BNN-Lasso | 0.63 | 40.9% | 42 | - | - | - |
| | BNN-MC-Dropout | 0.67 | 32.3% | 12 | - | - | - |
| | BNN-SGHMC | 0.53 | 51.3% | 505 | (111, 837, 1538) | 0.219 | 1.00 |
| | BNN-SGHMC(first 200) | 2.75 | 54.6% | 61 | (13, 111, 150) | 0.213 | 0.91 |
| | SWAG | 0.53 | 48.2% | 97 | (98, 1021, 1489) | 1.010 | 4.39 |
| | BNN-RNS-HMC | 0.62 | 44.7% | 38 | (107, 925, 1520) | 2.815 | 12.24 |
| | BNN-pCN | 0.65 | 51.1% | 620 | (99, 1003, 1532) | 0.159 | 0.69 |
| | FBNN (pCN-SGHMC) | 0.52 | 32.2% | 68 | (91, 912, 1533) | 1.338 | 5.95 |
| | FBNN (pCN-pCN) | 0.65 | 24.5% | 67 | (105, 1087, 1540) | 1.567 | 6.86 |
| | FBNN (SGHMC-SGHMC) | 0.50 | 40.0% | 70 | (77, 806, 1536) | 1.100 | 4.78 |
| | FBNN (SGHMC-pCN) | 0.52 | 48.1% | 57 | (92, 897, 1536) | 1.614 | 7.33 |
| Boston | DNN | 3.21 | - | 14 | - | - | - |
| Housing | Ensemble-DNN | 3.17 | 72.1% | 74 | - | - | - |
| | BNN-VI | 7.60 | 83.7% | 85 | - | - | - |
| | BNN-Lasso | 6.20 | 79.2% | 68 | - | - | - |
| | BNN-MC-Dropout | 10.12 | 81.2% | 91 | - | - | - |
| | BNN-SGHMC | 3.83 | 75.3% | 888 | (76, 649, 1536) | 0.085 | 1.00 |
| | BNN-SGHMC(first 200) | 7.40 | 66.9% | 86 | (9, 87, 150) | 0.104 | 1.22 |
| | SWAG | 5.81 | 71.9% | 104 | (68, 724, 1532) | 0.653 | 7.22 |
| | BNN-RNS-HMC | 9.42 | 73.4% | 76 | (73, 1032, 1504) | 0.960 | 10.6 |
| | BNN-pCN | 3.25 | 79.3% | 901 | (76, 649, 1536) | 0.084 | 0.88 |
| | FBNN (pCN-SGHMC) | 4.16 | 41.7% | 186 | (71, 965, 1543) | 0.381 | 4.22 |
| | FBNN (pCN-pCN) | 3.81 | 47.1% | 186 | (80, 966, 1541) | 0.430 | 4.78 |
| | FBNN (SGHMC-SGHMC) | 4.15 | 48.9% | 94 | (69, 979, 1542) | 0.734 | 8.22 |
| | FBNN (SGHMC-pCN) | 3.82 | 81.1% | 91 | (93, 938, 1543) | 1.021 | 11.94 |
| Alzheimer | DNN | 0.49 | - | 326 | - | - | - |
| Dataset | Ensemble-DNN | 0.42 | 89.3% | 1597 | - | - | - |
| | BNN-VI | 0.53 | 87.6% | 341 | - | - | - |
| | BNN-Lasso | 0.52 | 83.5% | 561 | - | - | - |
| | BNN-MC-Dropout | 0.60 | 92.8% | 268 | - | - | - |
| | BNN-SGHMC | 0.49 | 91.6% | 6524 | (102, 973, 1376) | 0.015 | 1.00 |
| | BNN-SGHMC(first 200) | 3.17 | 72.7% | 641 | (7, 82, 150) | 0.011 | 0.69 |
| | SWAG | 0.74 | 89.3% | 1214 | (106, 1002, 1542) | 0.087 | 5.58 |
| | BNN-RNS-HMC | 0.61 | 92.4% | 7324 | (96, 892, 1531) | 0.013 | 0.83 |
| | BNN-pCN | 0.51 | 89.9% | 6212 | (120, 1092, 1448) | 0.019 | 1.23 |
| | FBNN (pCN-SGHMC) | 0.50 | 90.2% | 643 | (116, 994, 1504) | 0.180 | 11.53 |
| | FBNN (pCN-pCN) | 0.56 | 91.4% | 682 | (108, 998, 1498) | 0.171 | 10.93 |
| | FBNN (SGHMC-SGHMC) | 0.55 | 88.4% | 671 | (118, 1012, 1541) | 0.176 | 11.24 |
| | FBNN (SGHMC-pCN) | 0.48 | 91.6% | 632 | (149, 984, 1497) | 0.218 | 13.97 |
| YearPredictionMSD | DNN | 74.54 | - | 1569 | - | - | - |
| Dataset | Ensemble-DNN | 72.89 | 90.47% | 7929 | - | - | - |
| | BNN-VI | 78.21 | 88.43% | 2146 | - | - | - |
| | BNN-Lasso | 79.44 | 89.04% | 3243 | - | - | - |
| | BNN-MC-Dropout | 83.08 | 87.89% | 1287 | - | - | - |
| | BNN-SGHMC | 81.67 | 83.81% | 25533 | (122, 1005, 1540) | 0.004 | 1.00 |
| | BNN-SGHMC(first 200) | 105.92 | 94.13% | 2613 | (7, 84, 149) | 0.002 | 0.56 |
| | SWAG | 93.87 | 86.33% | 3841 | (113, 987, 1537) | 0.029 | 7.35 |
| | BNN-RNS-HMC | 92.82 | 80.19% | 17289 | (109, 925, 1563) | 0.006 | 1.57 |
| | BNN-pCN | 82.45 | 85.74% | 25655 | (124, 873, 1631) | 0.004 | 1.20 |
| | FBNN (pCN-SGHMC) | 74.92 | 88.73% | 2815 | (143, 1049, 1676) | 0.050 | 10.63 |
| | FBNN (pCN-pCN) | 74.03 | 90.45% | 3046 | (138, 992, 1618) | 0.045 | 9.48 |
| | FBNN (SGHMC-SGHMC) | 76.69 | 90.40% | 2974 | (146, 973, 1599) | 0.049 | 10.27 |
| | FBNN (SGHMC-pCN) | 73.41 | 92.23% | 2724 | (156, 934, 1608) | 0.057 | 11.98 |

Table A2: Performance of various deep learning methods based on classification problems.

| Dataset | Method | Acc | Time(s) | ESS(min,med,max) | minESS/s | spdup | ECE |
|---|---|---|---|---|---|---|---|
| Simulated | DNN | 96% | 4257 | - | - | - | - |
| Dataset | Ensemble-DNN | 97% | 17415 | - | - | - | 0.382 |
| | BNN-VI | 90% | 4275 | - | - | - | 0.399 |
| | BNN-Lasso | 92% | 3189 | - | - | - | 0.363 |
| | BNN-MC-Dropout | 87% | 2912 | - | - | - | 0.277 |
| | BNN-SGHMC | 95% | 43841 | (47, 212, 1459) | 0.001 | 1.00 | 0.471 |
| | BNN-SGHMC(first 200) | 83% | 4218 | (21, 59, 156) | 0.004 | 4.64 | 0.498 |
| | SWAG | 81% | 4731 | (81, 773, 1368) | 0.017 | 15.97 | 0.482 |
| | BNN-RNS-HMC | 69% | 1309 | (135, 1190, 1493) | 0.103 | 96.20 | 0.513 |
| | BNN-pCN | 91% | 49774 | (36, 207, 1417) | 0.001 | 0.67 | 0.475 |
| | FBNN (pCN-SGHMC) | 93% | 5179 | (134, 959, 1419) | 0.051 | 24.13 | 0.409 |
| | FBNN (pCN-pCN) | 91% | 4858 | (146, 921, 1412) | 0.058 | 28.03 | 0.423 |
| | FBNN (SGHMC-SGHMC) | 95% | 4517 | (149, 891, 1540) | 0.032 | 30.76 | 0.406 |
| | FBNN (SGHMC-pCN) | 96% | 4489 | (154, 911, 1602) | 0.070 | 32.00 | 0.396 |
| Adult | DNN | 85% | 426 | - | - | - | - |
| Dataset | Ensemble-DNN | 84% | 2153 | - | - | - | 0.556 |
| | BNN-VI | 80% | 562 | - | - | - | 0.642 |
| | BNN-Lasso | 83% | 256 | - | - | - | 0.631 |
| | BNN-MC-Dropout | 82% | 187 | - | - | - | 0.540 |
| | BNN-SGHMC | 83% | 5979 | (16, 202, 1520) | 0.002 | 1.00 | 0.574 |
| | BNN-SGHMC(first 200) | 78% | 581 | (1, 41, 148) | 0.002 | 0.95 | 0.594 |
| | SWAG | 79% | 1641 | (47, 912, 1532) | 0.028 | 10.70 | 0.668 |
| | BNN-RNS-HMC | 72% | 6110 | (89, 960, 1530) | 0.014 | 5.44 | 0.658 |
| | BNN-pCN | 83% | 6227 | (9, 117, 1518) | 0.001 | 0.54 | 0.616 |
| | FBNN (pCN-SGHMC) | 82% | 642 | (87, 892, 1539) | 0.135 | 50.63 | 0.580 |
| | FBNN (pCN-pCN) | 82% | 639 | (88, 890, 1540) | 0.137 | 51.46 | 0.592 |
| | FBNN (SGHMC-SGHMC) | 83% | 612 | (68, 941, 1541) | 0.111 | 41.52 | 0.583 |
| | FBNN (SGHMC-pCN) | 84% | 609 | (89, 875, 1539) | 0.146 | 54.91 | 0.576 |
| Alzheimer | DNN | 82% | 51 | - | - | - | - |
| Dataset | Ensemble-DNN | 83% | 262 | - | - | - | 0.542 |
| | BNN-VI | 72% | 61 | - | - | - | 0.546 |
| | BNN-Lasso | 76% | 256 | - | - | - | 0.524 |
| | BNN-MC-Dropout | 76% | 12 | - | - | - | 0.429 |
| | BNN-SGHMC | 81% | 2736 | (81, 588, 1526) | 0.029 | 1.00 | 0.499 |
| | BNN-SGHMC(first 200) | 69% | 282 | (8, 84, 149) | 0.028 | 0.96 | 0.523 |
| | SWAG | 81% | 312 | (72, 913, 1562) | 0.231 | 7.69 | 0.508 |
| | BNN-RNS-HMC | 58% | 293 | (84, 915, 1540) | 0.286 | 9.55 | 0.521 |
| | BNN-pCN | 73% | 2660 | (71, 424, 1534) | 0.026 | 0.90 | 0.469 |
| | FBNN (pCN-SGHMC) | 76% | 277 | (76, 947, 1542) | 0.274 | 9.26 | 0.568 |
| | FBNN (pCN-pCN) | 77% | 274 | (70, 931, 1542) | 0.255 | 8.33 | 0.377 |
| | FBNN (SGHMC-SGHMC) | 80% | 278 | (81, 973, 1538) | 0.291 | 8.63 | 0.448 |
| | FBNN (SGHMC-pCN) | 84% | 280 | (92, 914, 1535) | 0.328 | 11.09 | 0.376 |
| Mnist | DNN | 92% | 231 | - | - | - | - |
| Dataset | Ensemble-DNN | 94% | 1129 | - | - | - | 0.312 |
| | BNN-VI | 86% | 273 | - | - | - | 0.417 |
| | BNN-Lasso | 87% | 184 | - | - | - | 0.445 |
| | BNN-MC-Dropout | 89% | 212 | - | - | - | 0.328 |
| | BNN-SGHMC | 90% | 8641 | (15, 364, 1456) | 0.001 | 1.00 | 0.280 |
| | BNN-SGHMC(first 200) | 78% | 916 | (1, 34, 149) | 0.001 | 0.62 | 0.271 |
| | SWAG | 83% | 1294 | (15, 431, 1376) | 0.011 | 7.72 | 0.327 |
| | BNN-RNS-HMC | 69% | 4541 | (14, 372, 1394) | 0.003 | 2.05 | 0.349 |
| | BNN-pCN | 88% | 8912 | (17, 398, 1471) | 0.001 | 1.26 | 0.321 |
| | FBNN (pCN-SGHMC) | 88% | 927 | (15, 412, 1383) | 0.016 | 10.78 | 0.352 |
| | FBNN (pCN-pCN) | 90% | 931 | (16, 393, 1421) | 0.017 | 11.45 | 0.312 |
| | FBNN (SGHMC-SGHMC) | 91% | 923 | (18, 409, 1461) | 0.019 | 13.01 | 0.283 |
| | FBNN (SGHMC-pCN) | 94% | 914 | (23, 474, 1521) | 0.025 | 16.77 | 0.241 |
| celebA | DNN | 80% | 3689 | - | - | - | - |
| Dataset | Ensemble-DNN | 82% | 15445 | - | - | - | 0.569 |
| | BNN-VI | 80% | 1132 | - | - | - | 0.622 |
| | BNN-Lasso | 79% | 2159 | - | - | - | 0.561 |
| | BNN-MC-Dropout | 78% | 1641 | - | - | - | 0.512 |
| | BNN-SGHMC | 81% | 17234 | (19, 383, 1537) | 0.001 | 1.00 | 0.567 |
| | BNN-SGHMC(first 200) | 69% | 1849 | (1, 85, 149) | 0.001 | 0.63 | 0.642 |
| | SWAG | 70% | 8913 | (85, 1014, 1467) | 0.009 | 8.65 | 0.534 |
| | BNN-RNS-HMC | 72% | 1835 | (72, 951, 1494) | 0.039 | 35.59 | 0.612 |
| | BNN-pCN | 76% | 19676 | (19, 331, 1538) | 0.001 | 0.95 | 0.529 |
| | FBNN (pCN-SGHMC) | 80% | 1972 | (99, 1155, 1542) | 0.050 | 45.53 | 0.565 |
| | FBNN (pCN-pCN) | 79% | 1951 | (116, 1155, 1542) | 0.059 | 53.93 | 0.542 |
| | FBNN (SGHMC-SGHMC) | 81% | 1904 | (93, 978, 1544) | 0.048 | 44.30 | 0.568 |
| | FBNN (SGHMC-pCN) | 85% | 1892 | (129, 785, 1517) | 0.068 | 61.84 | 0.493 |
| SVHN | DNN | 96% | 3748 | - | - | - | - |
| Dataset | Ensemble-DNN | 97% | 17415 | - | - | - | 0.382 |
| | BNN-VI | 90% | 4275 | - | - | - | 0.399 |
| | BNN-Lasso | 92% | 3189 | - | - | - | 0.363 |
| | BNN-MC-Dropout | 87% | 2912 | - | - | - | 0.277 |
| | BNN-SGHMC | 91% | 18639 | (31, 379, 1528) | 0.001 | 1.00 | 0.221 |
| | BNN-SGHMC(first 200) | 74% | 2515 | (5, 67, 262) | 0.001 | 1.19 | 0.246 |
| | SWAG | 83% | 6294 | (78, 499, 1383) | 0.012 | 7.45 | 0.282 |
| | BNN-RNS-HMC | 71% | 9083 | (19, 410, 1384) | 0.002 | 1.25 | 0.299 |
| | BNN-pCN | 93% | 17812 | (27, 398, 1317) | 0.001 | 0.91 | 0.275 |
| | FBNN (pCN-SGHMC) | 93% | 1931 | (36, 391, 1403) | 0.018 | 11.21 | 0.263 |
| | FBNN (pCN-pCN) | 91% | 1927 | (29, 372, 1280) | 0.015 | 9.04 | 0.318 |
| | FBNN (SGHMC-SGHMC) | 96% | 1891 | (34, 412, 1464) | 0.017 | 10.81 | 0.248 |
| | FBNN (SGHMC-pCN) | 96% | 1884 | (47, 486, 1533) | 0.024 | 14.99 | 0.223 |

