# OpenReview forum: "Scaling Up Bayesian Neural Networks with Neural Networks"
_TMLR — Accepted by TMLR_

### Review · Reviewer_Qk3U · 2024-06-07

**Summary Of Contributions:**

In this paper, the authors of the paper proposes a method to accelerate Markov Chain Monte Carlo based Bayesian neural networks (BNNs). The authors introduce a three-step Calibration-Emulation-Sampling (CES) strategy. First, a calibration step is conducted, where the authors of the paper use a standard stochastic gradient MCMC method to gather posterior samples from the BNN's parameter space. These samples enable us to train an emulator, or a deterministic neural network that approximates the likelihood computation of the stochastic BNN, which can be computationally expensive. With the emulator in hand, we can then speed up the BNN. Experimental results demonstrate that the CES approach significantly speeds up the process compared to traditional stochastic gradient MCMC methods.

**Audience:**

Yes

**Claims And Evidence:**

Yes

**Requested Changes:**

- It would be interesting to look at how accurate the emulator prediction is empirically.
- What is the architecture used for DNN/BNN and the emulator? How many parameters are there? How is the DNN trained? Such experimental details are important for reproducibility.
- The accuracy achieved by DNNs for MNIST seems very low compared to other common implementations. Any explanation for this?

**Strengths And Weaknesses:**

Strengths:
- The paper is clearly written in general.
- The proposed idea seems technically sound.

Weaknesses:
- While the main selling point of the paper seems to be on scaling and speeding up BNNs, the experiments were conducted on very small scale datasets.
- Some of the benchmark methods like MC dropout and SWAG can be applied to very large scale datasets with very deep neural networks. Based on my understanding, with the use of emulator, it can be hard to apply the proposed strategy to larger scale datasets. As such, I am not sure how much of a benefit the proposed speed up can bring in practice.

---

> ### Author Response · Authors · 2024-07-04
>
> Thank you for offering positive and constructive feedback. We have carefully reviewed each comment and provided detailed responses below, taking them into account.
>
> > Comment: While the main selling point of the paper seems to be on scaling and speeding up BNNs, the experiments were conducted on very small scale datasets. Some of the benchmark methods like MC dropout and SWAG can be applied to very large scale datasets with very deep neural networks. Based on my understanding, with the use of emulator, it can be hard to apply the proposed strategy to larger scale datasets. As such, I am not sure how much of a benefit the proposed speed up can bring in practice.
>
> Following the first set of reviews, we included much larger examples in our numerical experiments demonstrating that our results still hold. Based on your comment, we have expanded our evaluation to include even larger datasets. These include the Street View House Numbers (SVHN) dataset, with 630,420 labelled real-world images of house numbers and 3,072 features taken from Google Street View, and new simulation studies with 5,000,000 data points and 1,000 features. As you can see, our method still provides the best overall performance. We believe these new results further strengthen our confidence in the robustness and scalability of our approach, demonstrating its superiority in diverse scenarios and applications, while acknowledging that the method could be further improved in future research to handle even larger examples.
>
>
> > Comment: It would be interesting to look at how accurate the emulator prediction is empirically.
>
> Thank you for this great suggestion. We have included a new figure (Figure 5) that illustrates the accuracy of the emulator. In this figure, we compare the true log-likelihood function with its approximation from the emulator. As you can see, the results are very similar, indicating that the emulator provides a reasonable approximation of the target distribution.
>
>
> > Comment: What is the architecture used for DNN/BNN and the emulator? How many parameters are there? How is the DNN trained? Such experimental details are important for reproducibility.
>
>
> A new table (Table 2) has been included in the manuscript, outlining the architecture of the DNN emulator. This table specifies the number of hidden layers, the number of neurons per layer, the activation functions, the number of training epochs, and the dropout rates applied to the input layer and the first hidden layer.
>
>
> > Comment: The accuracy achieved by DNNs for MNIST seems very low compared to other common implementations. Any explanation for this?
>
> The reported accuracy achieved by DNNs for the MNIST dataset in our paper is specifically related to a binary classification task where the goal is to determine whether the label is digit 8 or not. This differs from the typical multi-class classification tasks for MNIST, which involve 10 labels (digits 0 through 9) and generally achieve a higher accuracy. Other studies in the literature that have used the same binary classification task have reported similar results: https://www.kaggle.com/code/lailaelmahmoudi123/binary-classification-for-the-mnist-dataset.

---

### Review · Reviewer_Szsx · 2024-06-16

**Summary Of Contributions:**

The paper proposed a novel method to enhance the computational efficiency of Bayesian Neural Networks (BNNs) by Calibration-Emulation-Sampling (CES) strategy. The authors introduced a Fast Bayesian Neural Network that leverages Deep Neural Networks (DNN) for the emulation component. Their methodology includes a preconditioning Crank-Nicolson (pCN) algorithm for efficient sampling from posterior distributions. The paper demonstrates the effectiveness of FBNN through extensive experiments on both simulated and real datasets, showing enhanced computational efficiency while maintaining prediction accuracy and robust uncertainty quantification.

**Audience:**

Yes

**Claims And Evidence:**

Yes

**Requested Changes:**

I suggest the author provide more details for the method section, and ideally make some illustrations to support the text.

**Strengths And Weaknesses:**

Strength:

- The paper provided extensive experimental results on both simulated and real datasets in this revision and included metrics for both calibration and accuracy.

- The proposed method addressed an important problem of BNN by speeding up the computation.

Weakness:

-  The datasets used are still cross-sectional data with low-dimensional features. Instead of BNN, classic machine learning models should also perform well.

- The definition of speed up is weird by comparing the effective sample size per second. How should it be interpreted as training / inference efficiency? Why deep ensemble is not compared in the speed up aspect.

- The description of the method is sketchy. Especially, it is unclear how the emulation step was conducted.

---

> ### Author Response · Authors · 2024-07-04
>
> We thank the reviewer for carefully reading our paper and providing constructive feedback. We have addressed each of your comments in detail below.
>
>
> > Comment: The datasets used are still cross-sectional data with low-dimensional features. Instead of BNN, classic machine learning models should also perform well.
>
> Thank you for your comment. Our goal is to provide a framework for uncertainty quantification with deep neural networks. Following the first set of reviews, we included much larger examples in our numerical experiments demonstrating that our results still hold. Based on the comments we received in this round, we have expanded our evaluation to include even larger datasets. These include the Street View House Numbers (SVHN) dataset, with 630,420 labelled real-world images of house numbers and 3,072 features taken from Google Street View, and new simulation studies with 5,000,000 data points and 1,000 features. As you can see, our method still provides the best overall performance, while allowing for uncertainty quantification. We believe these new results further strengthen our confidence in the robustness and scalability of our approach, demonstrating its superiority in diverse scenarios and applications, while acknowledging that the method could be further improved in future research to handle even larger examples.
>
>
>
> > Comment: The definition of speed up is weird by comparing the effective sample size per second. How should it be interpreted as training / inference efficiency? Why deep ensemble is not compared in the speed up aspect.
>
> ESS (Effective Sample Size) measures the number of ``independent'' samples in an MCMC chain, accounting for autocorrelation among the samples. When building a Markov chain, we calculate the ESS value for each parameter in the model to measure the amount of useful information in the collected samples for each model parameter. We report the Minimum ESS among  all the ESS values (for all model parameters). This indicates the worst case scenario, i.e., the slowest updates among all parameters.
>
> The speedup metric "spdup" evaluates the efficiency of each model by comparing their Minimum ESS per second to that of BNN-SGHMC as the benchmark. This metric shows how much faster or slower each model is in collecting effective samples compared to the baseline BNN-SGHMC method. Interpreting ESS in terms of computational efficiency, a higher ESS per second indicates that the model can generate more independent and useful samples in a given amount of computational time compared to baseline model, thus implying more efficient training.
>
> Deep ensembles were not compared in terms of speedup because they are not sampling methods, and thus, calculating ESS and the associated speedup metric is not applicable to them. In other words, deep ensembles use deterministic methods to combine multiple models' outputs, which doesn't involve generating samples in the same way as MCMC-based methods.
>
>
> > Comment: The description of the method is sketchy. Especially, it is unclear how the emulation step was conducted. I suggest the author provide more details for the method section, and ideally make some illustrations to support the text.
>
> Thank you for pointing this out. A detailed table (Table 2) has been included in the manuscript, outlining the architecture of the DNN Emulator. This table specifies the number of hidden layers, the number of neurons per layer, the activation functions used, the number of training epochs, and the dropout rates applied to the input layer and the first hidden layer. Moreover, we have included a new figure (Figure 5) that illustrates the accuracy of the emulator. In this figure, we compare the true log-likelihood function with its approximation from the emulator. As you can, the results are very similar, indicating that the emulator provides a reasonable approximation of the target distribution.

---

### Review · Reviewer_wiy7 · 2024-06-22

**Summary Of Contributions:**

**TL;DR**: this paper aims to improve the computational efficiency of Bayesian neural networks (BNNs), using an efficient implementation of calibration-emulation-sampling (CES). The proposed method improves efficiency while preserving the performance.

Specifically, each step of CES is modified as follows for improved efficiency:
- Calibration: the paper uses the stochastic gradient Hamilton Monte Carlo (SGHMC) algorithm to collect posterior samples for emulator training. The SGHMC is run only for a limited number of iterations (i.e. early-stopped) for efficiency.
- Emulation: The paper uses Fast Bayesian Neural Network (FBNN), a lightweight NN, as the emulator that maps from parameters to probabilities. The training data for FBNN is collected from the original Bayesian NN which is expensive to evaluate.
- Sampling: the paper chooses to use preconditioned Crank-Nicolson (pCN), which minimizes the correlations between samples and efficiently traverses the parameter space.

For theory, the paper shows that neural network emulators with size $\tilde{O}(\epsilon^{-d/(n-1)})$ that can the likelihoods and posteriors up to $\epsilon$ error in Hellinger distance.
- Here $d$ is the dimension of the parameter vector, $n$ is an upper bound on the parameter norm.

Empirically, the paper verifies the effectiveness of the proposed method on 10 synthetic and real-world datasets with both regression and classification tasks.
- Regarding efficiency, the original BNN requires 2000 samples, whereas FBNN can achieve comparable performance using only 200 samples. Overall, the proposed FBNN (SGHMC-pCN) has a 7 to 60 times speedup over the baseline BNN-pCN.
- The paper also compares with different variants where SGHMC and pCN are used at different steps of CES, and finds that the proposed method (i.e. SGHMC for calibration and pCN or sampling) is the most efficient.

**Audience:**

Yes

**Broader Impact Concerns:**

There are no direct ethical implications.

**Claims And Evidence:**

Yes

**Requested Changes:**

Please consider experiments with larger scale and higher dimensional data, or have a more thorough discussions on the difficulty of scaling up, such as the expected scaling behaviors.

For writing, it would be better if the presentation could be more concise, such as summarizing the main findings from the experiments.

**Strengths And Weaknesses:**

Strengths:
- The paper provides a method that practically improves the efficiency of an uncertainty quantification method, while preserving the performance.
- The paper is self-contained and provides the necessary background for technique involved in the proposed method.

Weaknesses: I am mainly concerned about the amount of contributions: the proposed FBNN (SGHMC-pCN) is modified based on the CES framework in prior work. The derivations for the theorems are standard, and the experiments are limited to small-scale datasets only, which the paper also acknowledges.

Some questions please:
- eq (3): where is $\mathcal{G}_n$ defined?
- "It should be noted that not all neural networks that represent uncertainty are Bayesian or even stochastic; some employ deterministic methods to estimate uncertainty without relying on stochastic components or Bayesian inference." -- please add references.
- Sec 3.2, calibration: could you justify why early stopping?
- Sec 3.5: $\alpha$ seems to be a $d$-dimensional vector; if that's the case, what is $|\cdot|$ in $\|\alpha|$?
- Sec 4.1: Why are BNN methods trained from a randomly chosen initial point for MCMC, whereas FBNN methods use warm starts? Is this unfair comparison, or is there some reason why warm starts can't be used for BNN?

---

> ### Author Response · Authors · 2024-07-04
>
> We thank the reviewer for valuable review of our submission and thoughtful feedback and suggestions. Below, we reply to the concerns and questions raised by the reviewer.
>
> > Comment: I am mainly concerned about the amount of contributions: the proposed FBNN (SGHMC-pCN) is modified based on the CES framework in prior work. The derivations for the theorems are standard, and the experiments are limited to small-scale datasets only, which the paper also acknowledges. Please consider experiments with larger scale and higher dimensional data, or have a more thorough discussions on the difficulty of scaling up, such as the expected scaling behaviors.
>
> Thank you for your comment. We addressed some of these issues after the first round of reviews (this is the second round, but new reviewers). We mentioned that the novelty of this paper is not developing a new approach; rather, making an existing approach work for a completely different and more challenging application. Our paper introduces a significant improvement in computational efficiency by integrating the CES method within a BNN framework. This adaptation required substantial modifications to the original CES to make it compatible and effective within the probabilistic modeling environment of BNNs. Also, following the first set of reviews, we included much larger examples in our numerical experiments demonstrating that our results still hold.
>
> Based on the comments we received in this round, we have expanded our evaluation to include even larger datasets. These include the Street View House Numbers (SVHN) dataset, with 630,420 labelled real-world images of house numbers and 3,072 features taken from Google Street View, and new simulation studies with 5,000,000 data points and 1,000 features. As you can see, our method still provides the best overall performance, while allowing for uncertainty quantification. We believe these new results further strengthen our confidence in the robustness and scalability of our approach, demonstrating its superiority in diverse scenarios and applications, while acknowledging that the method could be further improved in future research to handle even larger examples.
>
> > Comment: eq (3): where is $\mathcal{G}_n$ defined?
>
> Thank you for pointing this out. We made the necessary changes in manuscript to make this part more clear.
>
>
> > Comment: "It should be noted that not all neural networks that represent uncertainty are Bayesian or even stochastic; some employ deterministic methods to estimate uncertainty without relying on stochastic components or Bayesian inference." -- please add references.
>
> Done!
>
>
> > Comment: Sec 3.2, calibration: could you justify why early stopping?
>
> Early stopping is employed during the calibration step because the key focus at this step is not to achieve a precise approximation of the target posterior distribution. Instead, the focus is on collecting a small but representative set of training samples (from the parameter space) for the subsequent emulation step.
> By stopping early, we ensure that we collect a sufficient yet manageable set of model parameters.
>
>
> > Comment: Sec 3.5: $\alpha$ seems to be a $d$-dimensional vector; if that's the case, what is $|\cdot|$ in $|\alpha|$?
>
>
>  The notation $|\alpha|$ refers to the sum of the components of this multi-index. Specifically, if $\alpha = ( \alpha_1 , \alpha_2 , \cdots , \alpha_d)$, then:
>
> $|\alpha| =  \alpha_1 + \alpha_2 + \cdots + \alpha_d$
>
> This is a common notation used in the definition of Sobolev spaces, where $D^\alpha$ represents the mixed partial derivative of order $|\alpha|$. Here, $D^\alpha f$ would be the $\alpha_1$-th partial derivative with respect to the first variable, the $\alpha_2$-th partial derivative with respect to the second variable, and so on.
>
>
>
> > Comment: Sec 4.1: Why are BNN methods trained from a randomly chosen initial point for MCMC, whereas FBNN methods use warm starts? Is this unfair comparison, or is there some reason why warm starts can't be used for BNN?
>
>
> Utilizing these pre-collected samples as the initial point for subsequent sampling leverages the computational effort already expended and eliminates the need for additional random initialization. However, to ensure fair comparison, the computational cost of the warm start was taken into account when calculating the overall time to run each model.
>
>
>
> > Comment: For writing, it would be better if the presentation could be more concise, such as summarizing the main findings from the experiments.
>
> Thank you for the suggestion. We have removed some redundant information from the Numerical Experiment section to improve the presentation, and added a short paragraph to the Conclusion section briefly summarizing our findings.

---

### Comment · Action_Editor_2bNP · 2024-04-29
**This is a resubmission**

Dear reviewers,

This is a resubmission of the rejected manuscript:
https://openreview.net/forum?id=LjeF7DDm8Z
and the authors provided "Changes Since Last Submission" above.

Optimally, I'd assign the paper to the original reviewers, but all of them are unavailable unfortunately.

Best,
AC

---

### Author Response · Authors · 2024-07-04

We would like to thank the reviewers for their insightful feedback on our work. Below, we respond to specific questions and comments raised by the reviewers. We will also upload a revised version of our manuscript, with major changes highlighted in red.

---

### Decision · Action_Editor_2bNP · 2024-09-15

**Recommendation:** Accept as is

**Comment:**

The authors addressed the reviewers' concerns, and all reviewers found that the paper satisfies both criteria for TMLR publication.

**Audience:**

yes

**Claims And Evidence:**

yes